# Improving the fast-charging capability of NbWO-based Li-ion batteries

Yaqing Guo [1,7], Chi Guo[2,7], Penghui Li[3,7], Wenjun Song[1], Weiyuan Huang [4], Junxin Yan[3], Xiaobin Liao [5], Kun He [1] ✉, Wuxin Sha[6], Xuemei Zeng[1], Xinyue Tang[1], QingQing Ren[1], Shun Wang [1], Khalil Amine [4], Anmin Nie [3] ✉, Tongchao Liu [4] ✉ & Yifei Yuan [1] ✉

The discovery of Nb-W-O materials years ago marks the milestone of charging a lithium-ion battery in minutes. Nevertheless, for many applications, charging lithium-ion battery within one minute is urgently demanded, the bottleneck of which largely lies in the lack of fundamental understanding of $Li^+$ storage mechanisms in these materials. Herein, by visualizing $Li^+$ intercalated into representative $Nb_{16}W_5O_{55}$, we find that the fast-charging nature of such material originates from an interesting rate-dependent lattice relaxation process associated with the Jahn-Teller effect. Furthermore, in situ electron microscopy further reveals a directional, [010]-preferred $Li^+$ transport mechanism in $Nb_{16}W_5O_{55}$ crystals being the "bottleneck" toward fast charging that deprives the entry of any desolvated $Li^+$ through the prevailing non-(010) surfaces. Hence, we propose a machine learning-assisted interface engineering strategy to swiftly collect desolvated $Li^+$ and relocate them to (010) surfaces for their fast intercalation. As a result, a capacity of ≈ 116 mAh g$^{-1}$ (68.5% of the theoretical capacity) at 80 C (45 s) is achieved when coupled with a Li negative electrode.

The development of electric vehicles (EVs) is significantly influenced by advancements in fast-charging technology, which addresses key consumer concerns such as range anxiety and physical size and weight of batteries[1–3]. From a materials perspective, the diffusion of lithium-ion ($Li^+$) within the lattice structures is a critical bottleneck for fast charging[4]. This issue is exacerbated at the negative electrode of lithium-ion batteries (LIB), where fast charging can lead to overpotentials and safety concerns[5]. Graphite and silicon, both used as LIB negative electrodes, can store significant amounts of $Li^+$ within a potential range close to the $Li^+$/Li redox couple[6,7]. However, in practice, graphite stores $Li^+$ at a low potential close to the Li plating reaction, which can easily trigger lithium dendrite formation at high rates[8–10].

While silicon offers a higher theoretical capacity, its potential for fast charging remains uncertain due to its low electronic conductivity and substantial volume changes during cycling[11]. Beyond material selection, enhancing rate performance often involves creating nano or porous structures to increase the electrode-electrolyte interface contact, which, however, inevitably decreases its volumetric energy density[12–15]. It is thus crucial to explore fast-charging materials that not only exhibit high-rate performance but also maintain satisfactory energy density and safety[16].

Among many negative electrode candidates for fast charging, $Nb_2O_5$-$WO_3$ (NbWO) is particularly promising for its superior rate performance (up to 20 C)[17–19]. In addition to the high rate capability

[1]College of Chemistry and Materials Engineering, Wenzhou University, Wenzhou, China. [2]Jiangsu Key Laboratory for Design and Manufacture of Micro-Nano Biomedical Instruments, School of Mechanical Engineering, Southeast University, Nanjing, China. [3]Center for High-Pressure Science, State Key Laboratory of Metastable Materials Science and Technology, Yanshan University, Qinhuangdao, China. [4]Chemical Sciences and Engineering Division, Argonne National Laboratory, Lemont, IL, USA. [5]State Key Laboratory of Advanced Technology for Materials Synthesis and Processing, Wuhan University of Technology, Wuhan, China. [6]State Key Laboratory of Advanced Electromagnetic Engineering and Technology, School of Electrical and Electronic Engineering, Huazhong University of Science and Technology, Wuhan, China. [7]These authors contributed equally: Yaqing Guo, Chi Guo, Penghui Li. ✉e-mail: hekun@wzu.edu.cn; anmin@ysu.edu.cn; liut@anl.gov; yifeiyuan@wzu.edu.cn

of these materials, their voltage range of +3.0 to +1.0 V *vs.* Li[+]/Li also minimizes electrolyte decomposition and avoids lithium dendrite formation[20–23]. The crystalline structures of NbWO with different molar ratios of Nb-to-W can be divided into Wadsley-Roth shear structure (Nb:W > 2) and tetragonal tungsten bronze structure (Nb:W ≤ 2)[24,25]. $Nb_{16}W_5O_{55}$ is one typical representative of the former with blocks of corner-sharing octahedra of size $(4 \times 5)_1$ (Nb, W)$O_6$ octahedra, which are connected by edge-sharing (Nb, W)$O_4$ tetrahedra[26,27]. Though with a decent fast-charging performance, similar to other members of the NbWO family, $Nb_{16}W_5O_{55}$ shows poor capacity retention below 50 mAh g$^{-1}$ at fast-charging conditions such as 60 C (one-minute charging), the improvement of which is currently limited by the lack of in-depth understandings of its Li$^+$ storage mechanisms[5].

Theoretically, the charging mechanism of the $Nb_2O_5$-$WO_3$-based Wadsley-Roth shear structure has been investigated by density functional theory (DFT) calculations[28]. Andrew J. Morris et al. [27,29] found that tungsten primarily occupies tetrahedra and block-central sites within block-type crystal structures, influencing electrochemical performance through crystallographic shear planes and ReO$_3$-like block interiors. These features impede Li$^+$ movement through shear planes while promoting rapid diffusion through open tunnels, creating effective one-dimensional lithium diffusion pathways[30]. Beyond theoretical predictions, however, there are currently no further understandings from experimental points of view to clarify the underlying factors that plague their fast-charging future, not to mention any effective strategies to be practically implemented[31–33]. Therefore, further investigation into the fast-charging mechanism of NbWO and the performance limitations at ultra-high rates is essential. This will enhance its rapid charging capabilities and meet the demands of fast-charging applications.

In this work, through direct visualization of Li$^+$ electrochemically intercalated into micron-sized $Nb_{16}W_5O_{55}$, we discovered that the material's fast charging capability is derived from a rate-dependent lattice relaxation process associated with the Jahn-Teller effect. We observed that the NbWO structure is highly sensitive to the rate: lithium ions tend to occupy lattice sites randomly at higher rates, alleviating the significant lattice distortions inherent in the NbWO structure, and facilitating the rapid intercalation of lithium ions. Anomalously, at lower rates such as 0.1 C, lattice distortions became more pronounced, adversely affecting the structural stability. Moreover, [010]-preferred Li$^+$ transport in $Nb_{16}W_5O_{55}$ crystals is verified to be the major bottleneck for fast charging, during which the entry of any desolvated Li$^+$ into $Nb_{16}W_5O_{55}$ bulk through the prevailing non-(010) surfaces is deprived. In light of these mechanistic insights, we propose the machine learning-assisted interface engineering strategy, which acts as a Li-attracting interface to collect the desolvated Li$^+$ against re-solvation at the electrode/electrolyte interface and further transport them to the (010)-faceted entrance for swift Li$^+$ intercalation, largely alleviating the anisotropy aftermath and enhancing the charging rate. Hence, it resulted in a significant improvement in high-rate performance (≈116 mAh g$^{-1}$ at 80 C, 45 s), far surpassing traditional $Nb_{16}W_5O_{55}$ (which exhibits a capacity below 50 mAh g$^{-1}$ at 60 C, 60 s), and meets the fast-charging criteria outlined by USABC (United States Advanced Battery Consortium defines fast charging as reaching 80% state of charge within 15 min (4 C rate), and our engineered NbWO achieves this even at nearly 40 C, 90 s)[34]. The results obtained from this investigation will contribute to the knowledge of lithium storage mechanisms in fast-charging materials, contributing to the development of advanced fast-charging batteries.

## Results
### Crystal structure of $Nb_{16}W_5O_{55}$
Figure 1a schematically illustrates the crystal structure of $Nb_{16}W_5O_{55}$, derived from stepwise annealing. Initially, H-type $Nb_2O_5$ (H-$Nb_2O_5$)

crystals were synthesized via NbC precursor annealing at 950 °C for 10 h in air (Supplementary Fig. 1). Energy-dispersive X-ray spectroscopy (EDS) mappings confirm homogeneous Nb and O distribution in H-$Nb_2O_5$ (Supplementary Fig. 2a). This annealing process transforms irregularly shaped NbC into micrometer-scale H-$Nb_2O_5$ particles with 0.39 nm interplanar spacing corresponding to the ($\bar{4}05$) crystal plane (Supplementary Fig. 2b and c). The high-angle annular dark-field scanning transmission electron microscopy (HAADF STEM) imaging at [010] zone axis shows the block structure with aligned dots in 2 × 3 and 2 × 4 (Supplementary Fig. 3a), matching the modeled pattern (Supplementary Fig. 3b). The crystal comprises Nb-$O_4$ tetrahedra and Nb-$O_6$ octahedra; Nb-O bond lengths are nearly uniform, resulting in the orderly aligned oxygen atoms along the electron beam direction that strengthen the corresponding oxygen signal in the STEM-HAADF image (Supplementary Fig. 3c and d). Subsequently, annealing of the H-$Nb_2O_5$ in conjunction with $WO_3$ led to the formation of $Nb_{16}W_5O_{55}$ (NbWO) crystals. Scanning electron microscopy (SEM) images reveal that the $Nb_{16}W_5O_{55}$ particles uniformly exhibit stick-like morphologies and grow along the b-axis (Supplementary Fig. 4).

The crystal structure of $Nb_{16}W_5O_{55}$ at [010] zone axis comprises units made of (4 × 5) $MO_6$ octahedra (M = Nb, W, illustrated in the red block area of Fig. 1a) and tetrahedra at the corners of these blocks, each block contains 12 parallel tunnels on the a-c plane. These blocky units are interconnected along their edges by crystallographic shear planes, resulting in a noticeable shear structure[26]. The HAADF STEM images elucidate these tunnel structures, matching with the [010]-projected theoretical models shown in Fig. 1b. Furthermore, the fast Fourier transform (FFT) pattern, displayed in Fig. 1c along the [010] zone axis, reveals that the experimental and theoretical results are in good agreement[21]. Locally, due to the significant difference in the elemental periods of Nb and W and the irregular arrangement of cations, the crystal structure prominently displays distorted octahedra (Fig. 1a, the gray octahedra represent W-$O_6$, while the blue octahedra represent Nb-$O_6$, with Nb and W randomly occupying positions). This distortion arises from the interplay between electrostatic repulsion among cations and the second-order Jahn-Teller (SOJT) effect[29]. The NbO$_6$ octahedra situated at the edges of the blocks are subject to more pronounced distortion compared to those at the core, leading to formation of zigzag patterns of metal cations along the crystallographic shear planes, as shown in Fig. 1d under another zone axis for better illustration[29]. As shown in Fig. 1e, a crystallographic shear structure can be observed with an interplanar distance of 0.38 nm, consistent with the (11$\bar{1}$) crystal plane of $Nb_{16}W_5O_{55}$, and the crystal structure can be perfectly matched to the [101]-projected theoretical model together with the consistent FFT patterns (Fig. 1f). X-ray photoelectron spectroscopy (XPS) shows that Nb 3$d$ spectra can be divided into two peaks corresponding to Nb $3d_{5/2}$ (209.3 eV) and Nb $3d_{3/2}$ (206.5 eV), indicating the presence of Nb$^{5+}$ (Supplementary Fig. 5a). In addition, the W 4$f$ spectra exhibit two peaks at 37.1 eV and 34.9 eV, which can be respectively denoted as the W $4f_{5/2}$ and W $4f_{3/2}$ with ΔE around 2.2 eV, indicating a valence state of W$^{6+}$ (Supplementary Fig. 5b)[21]. These noticeable blocks and shear structures lay the foundation for exploring the fast-charging mechanism of this material, which will be the focus of our upcoming investigations.

### Fast charging mechanism of $Nb_{16}W_5O_{55}$ crystal
The reaction of $Nb_{16}W_5O_{55}$ with Li proceeds in three distinct regions from 3.0 V to 1.0 V in the CV curve (Supplementary Fig. 6a), as characterized by the slopes of the voltage profile (Supplementary Fig. 6b). The curve exhibits a noticeable peak occurring at 1.64 V during the lithiation process, primarily corresponding to the reduction of Nb$^{5+}$. Notably, NbWO demonstrates a high lithium-ion diffusion coefficient at 2.1 V and 1.7 V during the intercalation process of Li$^+$, as determined by the galvanostatic intermittent titration technique (GITT, illustrated in Supplementary Fig. 7). In addition, at a rate of 0.2 C, approximately

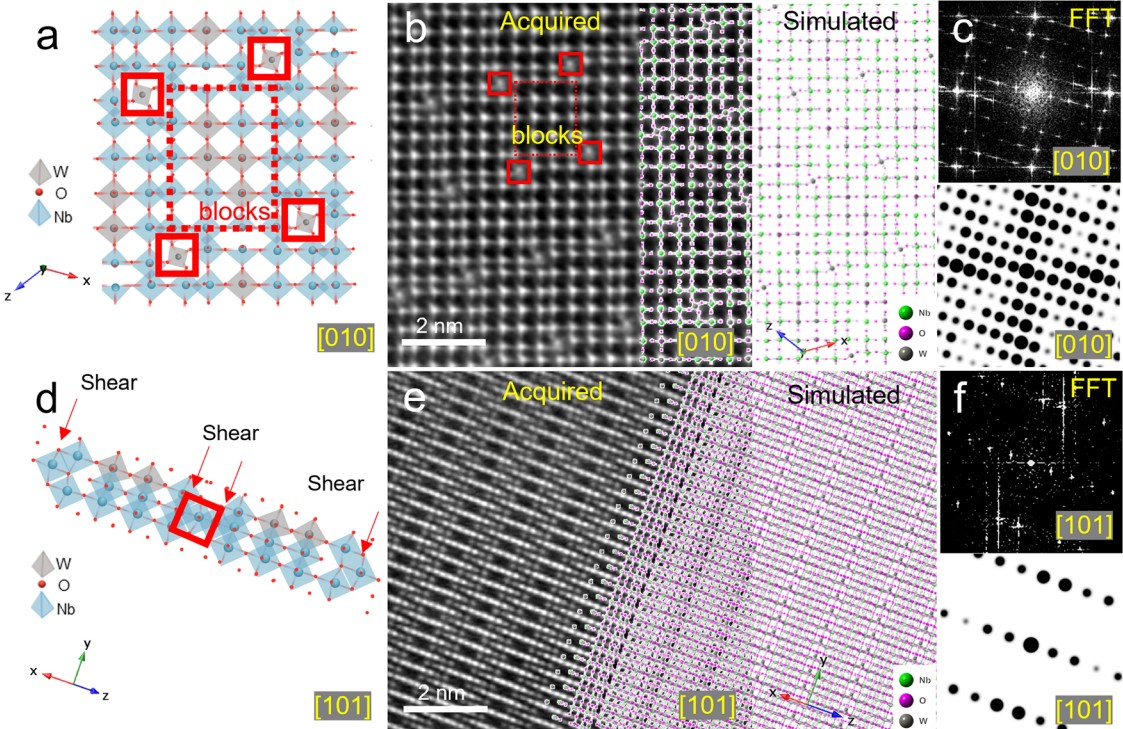

**Fig. 1 | Characterizations of Nb₁₆W₅O₅₅ nanocrystals. a–c** Schematic of crystal structure (**a**), atomic scale HAADF STEM images (**b**) of [010] zone axis for Nb₁₆W₅O₅₅, and the corresponding FFT patterns (**c**) along the tunnel directions, the atom arrangements (acquired) are in good agreement with their theoretical models (simulated) where green, grey, and purple represent Nb, W, and O, respectively. **d–f** Schematic of crystal structure (**d**), atomic scale HAADF STEM images (**e**), and the corresponding FFT patterns (**f**) of [101] zone axis for Nb₁₆W₅O₅₅.

1.3 $Li^+$ per transition metal ($Li^+$/TM) can be reversibly intercalated, resulting in a capacity of about 227 mAh g⁻¹ (Supplementary Fig. 6c). When the rate is increased to 5 C, $Nb_{16}W_5O_{55}$ maintains a capacity of 170 mAh g⁻¹. At this rate, the reaction plateau remains evident; however, when the rate is further increased to 20 C and beyond, which corresponds to a three-minute discharge, the slope of the curve significantly increases. This obvious change is likely related to alterations in the crystal structure of the material under different rates. Consequently, we conducted a detailed investigation into the crystal structure of the material at various rates.

In order to further explore the fast-charging mechanism of NbWO, we carried out atom-resolving STEM and density functional theory (DFT) analyses of NbWO crystals. The redox centers $Nb^{5+}$ and $W^{6+}$ in NbWO are located in six-coordinate environments. However, due to the significant difference in atom size, the distribution of Nb and W in the crystal exhibits a disordered but not random pattern[29]. As shown in Fig. 2a, b, the $M\text{-}O_6$ octahedra structure, where M represents Nb and W, exhibits distinct uneven bond length characteristics with shorter W-O bond lengths due to the electronegativity of the +6 oxidation state of tungsten. In addition, due to the combination of electrostatic repulsion between the cations and the irregular arrangement of M atoms (where M = Nb or W) in the M-O-M, the octahedra are severely distorted (Fig. 2a). As shown in annular bright field (ABF) reverse color image of Supplementary Fig. 8, where a dot represents a column of metal atoms, the outlines of the dots are more blurred and the lattice sites of O are ultimately not in a straight line due to the different bonding interactions between the transition metals (Nb and W) for O. As a result, the ABF image shows only Nb, W without O due to arrangement irregularities of O within their specific columns, which is quite different from the $H\text{-}Nb_2O_5$ lattice with homogeneous Nb-O bonds in $Nb\text{-}O_6$ octahedra where the sharp imaging signal of O can be clearly seen in the $H\text{-}Nb_2O_5$ (Supplementary Fig. 3). At the same time, it can be seen that the metal at the tetrahedra sites is more clearly

aligned with the O sites, indicating more orderly atomic arrangement in the tetrahedra region. As shown in Supplementary Fig. 8d, the atomic sites in the same column deviate from their central position, and those in the shear plane are more severely shifted, leading to significant lattice distortions. Combined with DFT calculations (Supplementary Fig. 9), it is found that when a lithium-ion is intercalated in the block, the occupation of lithium-ion will be different due to the disordering of the crystal structure. When the lattice atoms are more ordered, the lithium ions tend to occupy the center of the tunnel space; when the lattice distortion is severe, the lithium ions tend to occupy the off-center positions to reduce the total energy of the crystal structure.

To further investigate the $Li^+$ diffusion kinetics and thermodynamics of the NbWO crystal, detailed analyses of the crystal structure discharged to specific cut-off voltages under different current densities were carried out (for the preparation and testing methods of the Li ||NbWO half-cell, please refer to the electrochemical characterization section). To clearly delineate the fast-charging mechanism, we referred to the reverse-colored annular bright field (ABF) STEM method to enhance the visibility of $Li^+$ within the NbWO crystal, with the aim of identifying $Li^+$ storage sites (depicted as white sites in the tunnels in the reverse color image) within the tunnel structure. We compared the ABF STEM images of the crystals at charge rates of 0.1 C, 20 C, and 80 C, with discharge voltages of 2.1 V, 1.7 V and 1 V, as shown in Supplementary Figs. 10 and 11. Specifically, at a discharge rate of 0.1 C to 2.1 V, the crystal structure is identified as $Li_{1.7}Nb_{16}W_5O_{55}$ (Supplementary Fig. 12) with a low lithium content, allowing the crystal sufficient time to relax and maintain a thermodynamically stable state, despite the severe lattice distortions (as seen in Fig. 2c). At 1.7 V, with the intercalation of the $Li^+$ ions, the influence of the Li-O bond gradually increases, and the lattice arrangement shows localized order (Fig. 2d), diminishing distortions and increasing local octahedra symmetry with visible $Li^+$ sites and a transition from eccentric to central $Li^+$

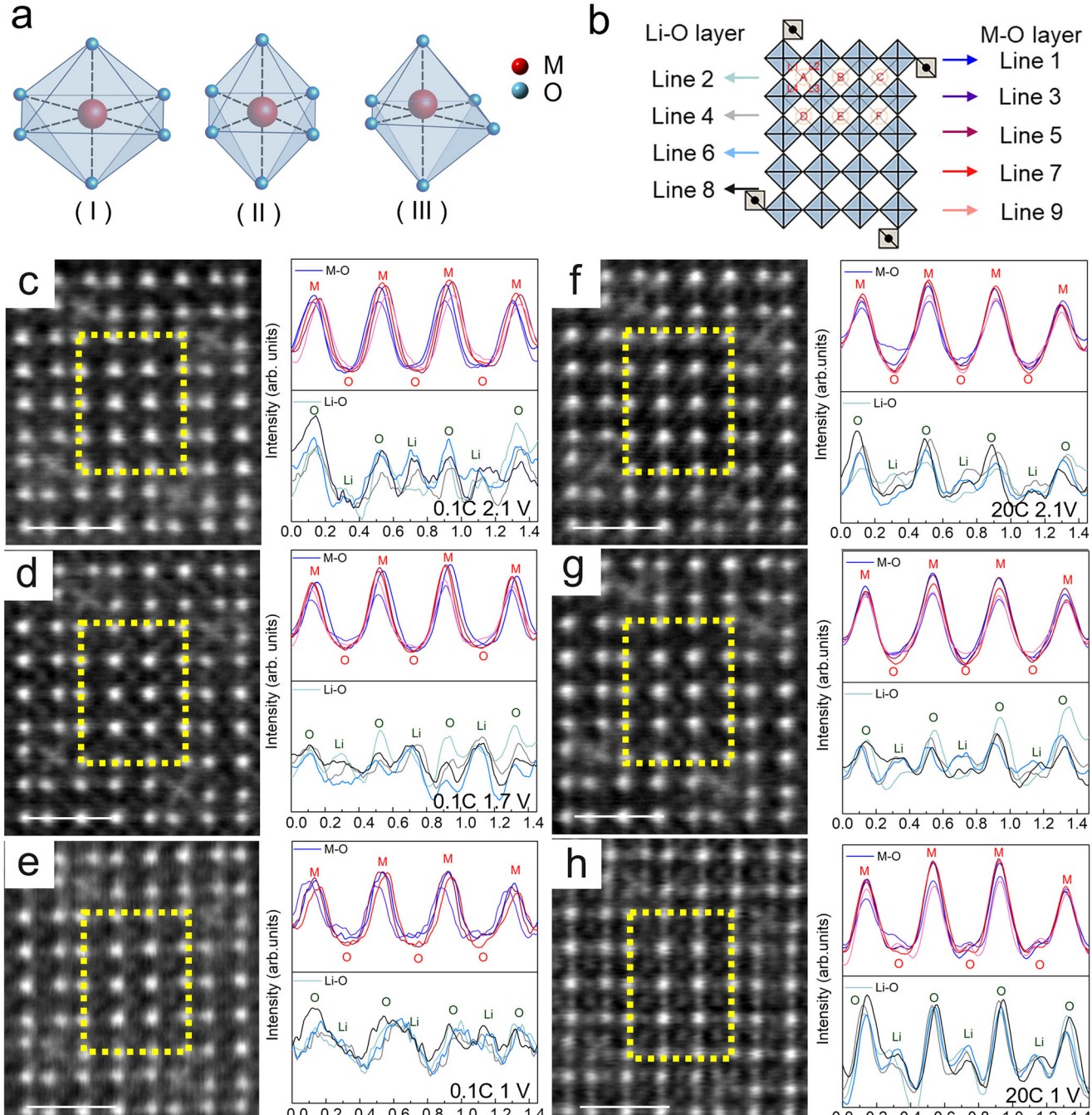

**Fig. 2 | Fast charging mechanism of Nb$_{16}$W$_5$O$_{55}$ crystal. a** The M-O$_6$ octahedra structure, where M represents niobium (Nb) and tungsten (W), I indicates that the six M-O bond lengths are identical, II represents the elongation of the two axial bonds in the M-O$_6$ octahedron, and III signifies a random distribution of the six M-O bond lengths with significant displacement of the M-site. **b** Schematic diagram of the lattice sites and Li sites of the block structure, where the red and blue represent the M and O sites, while the left arrow represents the Li and O sites. ABF

STEM images (the images are presented in reverse color) at [010] zone axis, and the intensity line profile is inverted and displayed as peaks along the arrows indicated in (**b**) for Nb$_{16}$W$_5$O$_{55}$ discharge with the rate of (**c**–**e**) 0.1 C and (**f**–**h**) 20 C at 2.1 V, 1.7 V, and 1 V, respectively, scale bar is 1 nm. The yellow boxes represent the (4 × 5) MO$_6$ blocks in the material. The Y-axis value for the basis line reflects the intensity of the darkest location within the area of interest, X-axis with the unit of nm.

occupancy, as detailed in the Supplementary Tables 1 and 2 (the errors between L1/L3 and L2/L4 are considered to be the center of the occupancy for an error of 0.3 Å, highlighted in Fig. 2b). Upon further intercalation of Li$^+$ (discharged to 1 V *vs*. Li/Li$^+$), the crystal structure undergoes collapse due to the intense electrostatic repulsion between Li$^+$ ions leading to a major distortion of the octahedra, as depicted in Fig. 2e and Supplementary Table 3. The results indicate that at 0.1 C, as Li$^+$ ions intercalate, the initially highly distorted NbWO crystal undergoes a lattice relaxation process due to the strengthening of Li-O bond

interaction, which leads to the recovery of the crystal ordering with suppressed distortion. However, with the accumulation of Li$^+$, the crystal exhibits thermodynamic instability, and the intercalation of a large amount of Li$^+$ can lead to crystal collapse.

During charge and discharge cycles, a complete discharge occurs within 3 min at a rate of 20 C. The crystal changes at this rate are different, and the distortion of the MO$_6$ octahedron is eliminated mainly at the beginning of the discharge (2.1 V). Due to the rapid lithiation speed, Li$^+$ does not intercalate based on the principle of

minimum energy but rather intercalates randomly (Supplementary Tables 4–6). This phenomenon leads to an enhanced interaction between oxygen in M-O bonds and Li$^+$ ions, particularly in regions with significant lattice distortion, where selective intercalation occurs at 0.1 C. Consequently, the distortion of M-O$_6$ octahedra becomes less, resulting in a more orderly arrangement of the M atomic columns (Fig. 2f). Li$^+$ ions intercalation into distorted tunnels leads to a relaxation of such lattice distortions at M and O sites due to enhanced Li-O coordination, which, in turn, facilitates rapid lithium ion intercalation. With the continuous intercalation of lithium ions, the Li$^+$ ion sites become distinctly visible (Fig. 2g), with complete alleviation of lattice distortion and orderly oxygen arrangement, making the O sites apparent in the ABF images at 1 V (Fig. 2h). Moreover, our investigations on NbWO at an high rate of 80 C further confirm that even at lower Li$^+$ ions concentrations, the rapid lithiation rate leads to random rather than thermodynamically favored lowest-energy intercalation at sites such as the 5-fold coordinated ones near the shear structure in NbWO crystal[29]. This random insertion mitigates crystal distortions, resulting in the oxygen lattice signals at 80 C being significantly pronounced compared to those at 0.1 C (Supplementary Fig. 11). Additionally, DFT calculations were used to simulate the voltage curves of NbWO at homogeneous and heterogeneous intercalation models. The results show that at a homogeneous intercalation model (corresponding to low discharging rates), each NbWO cavity is occupied by one Li$^+$ following the minimum energy principle, resulting in a distinct discharge plateau in the voltage curve, consistent well with the charge-discharge curves. However, in the heterogeneous intercalation model (corresponding to high discharging rates), due to the randomness of Li$^+$ occupancy, the calculated voltage fluctuates significantly and decays rapidly (Supplementary Fig. 13). Hence, the fast-charging potential of NbWO is thermodynamically controlled rather than kinetically driven, which could be achieved by the initial fast Li$^+$ intercalation step with the removal of crystal distortions that stabilizes the material in a high-energy state conducive to further rapid intercalation of Li$^+$ ions through regular tunnel sites. However, these discoveries are rooted in the inherent properties of the material, which can facilitate fast-charging capabilities. Yet, the factors limiting performance enhancement remain unexplored.

**Anisotropic lattice change response to lithium-ion intercalation**
Since our investigation focuses on the diffusion mechanism of lithium ions within the bulk of the material, in situ TEM was employed to observe the electrochemical lithiation process in NbWO to further investigate the limiting factors of NbWO's performance. This method has been widely used to study Li$^+$ storage mechanisms in various electrode materials with high reliability, such as the consistency between the lithiation kinetics of silicon negative electrode studied via in situ solid-state TEM, in situ liquid TEM and ex situ TEM of real battery-cycled Si negative electrode[35–37]. The single NbWO crystals for in situ TEM imaging under its [010] and [101] zone axes were obtained by focused ion beam (FIB) thinning along the cross-section ([010] zone axis) and perpendicular to the cross-section ([101] zone axis, Supplementary Fig. 14), respectively. The experiments were conducted using an all-solid electrochemical cell setup, comprising a working electrode of NbWO single crystal, a naturally grown Li$_2$O solid electrolyte, and a bulk lithium metal counter electrode (Fig. 3a). A bias potential of -3 V was used to initiate the lithiation of the bulk in NbWO particle.

Figure 3b presents a cross-sectional view along the [010] zone axis of NbWO, highlighting the intercalation of lithium ions from a side perpendicular to the [010] zone axis. The material surface exhibited no significant streaks at the start of the video (Supplementary movie 1 at 0 s). However, upon contact with Li$_2$O, the NbWO surface developed a stress streak that remained unchanged until a -3 V voltage was applied to facilitate Li$^+$ ions intercalation (Supplementary movie 1 at 200 s). Upon reaching 200 s of lithiation (equivalent to 400 s of video time),

the overall material showed no significant changes, with no noticeable expansion of the streak observed. Likewise, the crystal structure remains consistent at 800 s of lithiation (or 1000 s of video time). At this point, the Li$_2$O is positioned at the tungsten (W) tip, away from the area where compressive stress is released, thus minimizing the potential for displacement of the crystal streak. To explore the possible relationship between stripe displacement and changes in the crystal phase structure, selected areas of the material were subjected to the selected area electron diffraction (SAED) analysis. These areas included location 2, a flat region away from the lithiation site; location 3, at the site of lithiation; and location 4, within the streaked area but distant from the lithiation point. The inset in Fig. 3b shows the original SAED patterns, with interplanar spacings of 1.55 nm and 1.57 nm, matching the spacing of (200) and (001) crystal planes, respectively. After lithiation, there were no notable changes in the interplanar spacings at loci 2, 3, and 4. The minor variations observed in the SAED results are attributed to a slight shift in the crystallographic zone axis due to stress rather than changes in the spacing of the crystal planes. Subsequently, we zoomed in on the area of contact at site 3 in Fig. 3b, which was in close contact with Li$_2$O after 800 s of lithiation, and captured an ABF STEM image. As shown in Fig. 3c, due to the contact occurring through a shear plane perpendicular to the [010] zone axis, it was challenging for Li$^+$ to traverse the shear plane for lithiation. Consequently, no distinct features of lithium ions occupying the tunnel structures were observed in the ABF image.

To further explore the Li$^+$ ions intercalation mechanism, the lithium intercalation process along the b-axis of NbWO was documented using a TEM video (Fig. 3d and Supplementary movie 2). The initial TEM image at 0 s displays the crystal structure of NbWO, with SAED analysis confirming it to be a single crystal oriented along the [010] growth direction, and the lattice spacing of 1.49 nm pointing to the ($\bar{2}$01) crystal plane. At 140 s in the video, Li$_2$O contact with NbWO causes compressive stress to gather on the contact surface. The strain gathering remained unchanged, and subsequently, a voltage of −3 V was applied to facilitate the intercalation of lithium ions. Notably, after 17 s of lithium intercalation (equivalent to 157 s of video time), distinct streaks indicative of Li$^+$ ions intercalation become visible, extending along the b-axis by 23 s of lithium intercalation (equivalent to 163 s of video time). Driven by a solid-solution lithiation mechanism, the targeted particle exhibits a relatively small volumetric expansion (merely 5.5% at 1.0 Li$^+$/TM), and no significant volume expansion is seen in the low-magnification TEM images. Following the b-axis-directed lithiation, the crystal primarily expands along this axis, the trend of which is, however, physically restricted by two fixed metallic terminates (tungsten and copper) as evidenced by considerable strain concentration at 254 s (equivalent to 394 s of video time). Eventually, lithiated for 269 s (equivalent to 409 s of video time), the crystal fractures along the (010) plane due to stress overloading. Post-fracturing SAED analysis shows an increase in lattice spacing of ($\bar{2}$01) from 1.49 nm to 1.56 nm, which could be attributed to Li$^+$ ions intercalation. Despite the deflection of the crystal zone axis due to fracturing, the diffraction pattern indicates that no new phase emerges during the lithiation process. Furthermore, when Li$_2$O directly contacts the [010] zone axis, lithium ions can easily intercalate through the tunnels, resulting in an increased lattice spacing of ($\bar{2}$01). This change is evident in the cross-sectional images, where the accumulation of lithium ions within the tunnels can be clearly observed (Fig. 3e). Hence, as depicted in Fig. 3f, Li$^+$ ions can only undergo rapid intercalation and deintercalation through the [010]-oriented tunnels. However, in the stick-like NbWO elongated also along [010], the exposed surface of the particle is mostly the shear planes (Supplementary Fig. 4) instead of (010) preferred by Li$^+$ intercalation. This leads to sluggish Li$^+$ intercalation around these shear planes, resulting in accumulation of more Li$^+$ at these surface regions, which could be further re-solvated into the electrolyte with thus severely compromising rate capability. As such, realizing that the

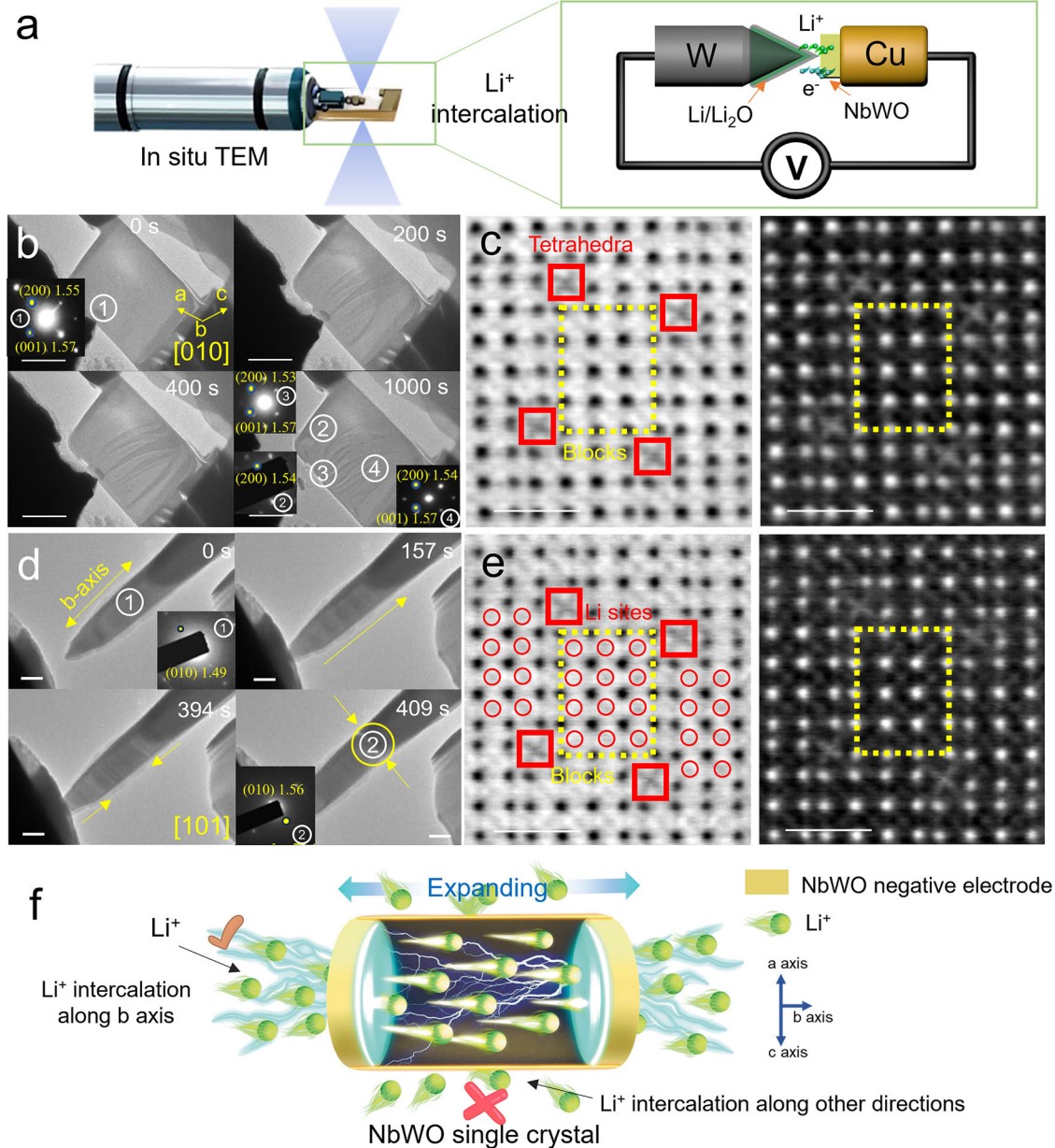

**Fig. 3 | Anisotropic lattice change upon lithium-ion intercalation. a** Schematic illustration of the battery setup for in situ TEM analyses. **b** Real-time lithiation behavior of NbWO crystal along the direction of lithium ions intercalated perpendicular to the [010] zone axis (scale bar: 500 nm). The inset of figures correspond to the SAED of the corresponding region in (**b**). **c** The ABF STEM image and the corresponding reverse-colored ABF STEM image, taken along the [010] zone axis, display the atomic structure of the region 3 after lithium intercalation for 1000 s (shown in (**b**)), the yellow boxes represent the (4 × 5) $MO_6$ blocks in the material.

**d** Real-time lithiation behavior of NbWO crystal along the [010] orientation (scale bar: 100 nm). The inset of figures correspond to the SAED of the corresponding region in (**d**). **e** The ABF STEM image and the corresponding reverse-colored ABF STEM image, taken along the [010] zone axis, display the atomic structure of the needle tip region after lithium intercalation for 409 s (shown in (**d**)). The yellow boxes represent the (4 × 5) $MO_6$ blocks in the material. **f** Schematic of anisotropic Li+ ions intercalation model.

anisotropic Li+ transport property of the NbWO host acts as a major bottleneck toward approaching its ultrafast-charging milestone within 1 min, we will discuss in the following how such anisotropy and its aftermath could be smartly addressed using a simple surface coating strategy.

## Fast charging performance enhancement and evaluation

As discussed above, we herein propose that the interface engineering strategy could function to relocate the desolvated lithium ions surrounding the NbWO surfaces to the openings of [010]-oriented tunnels where Li+ intercalation is more favorable with a low energy barrier, which can overcome the above-mentioned bottleneck for fast

charging in NbWO materials. Firstly, a high-throughput screening process is implemented to explore potential coating materials for the negative electrode, as shown in Fig. 4a. Around 83,989 structure and property data entries were obtained from the Materials Project database[38] (detailed calculation methods are provided in the Supplementary Information). Each entry contains data such as the crystal structure, space group number, formation energy, and band gap of the material. In lithium-ion batteries, the electrodes are mixed conductors that require both good electronic and ionic conductivity to connect with external and internal circuits. Therefore, the electrode coating materials also need to have high electronic and ionic conductivity, as well as a high oxidation potential to maintain electrochemical stability.

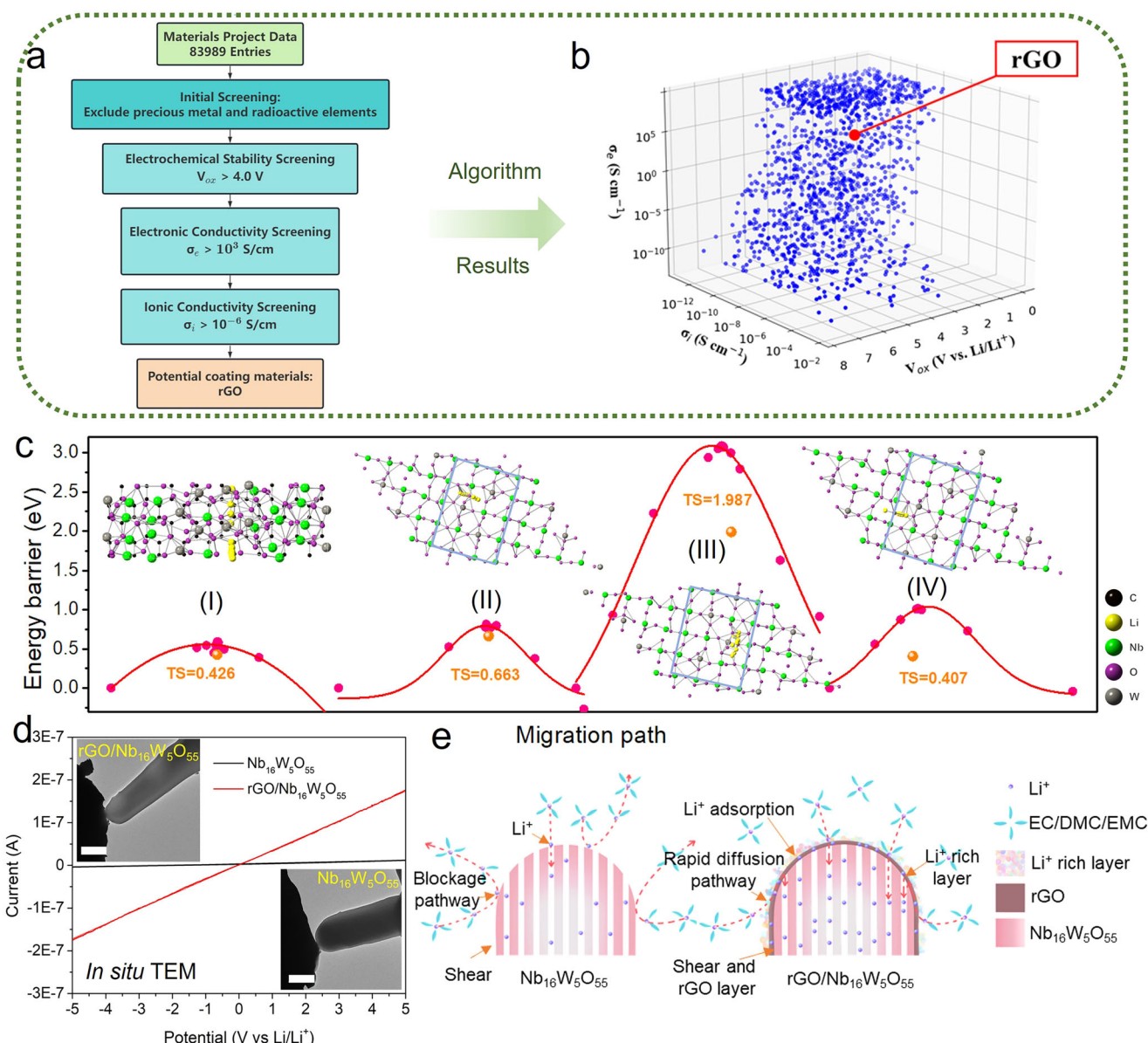

**Fig. 4 | Fast charging performance enhancement. a** The overall screening procedure from the Materials Project database. **b** Initial screening results of candidate coating materials under 3 selected dimensions: electronic conductivity, ionic conductivity, and oxidation potential. **c** Minimum energy paths and activation barriers (NEB) for lithium-ion motion in rGO/NbWO. **d** I-V curves of NbWO and rGO/NbWO particles measured in situ under the microscope (scale bar 1 μm). **e** Schematic illustrations of lithiation processes of NbWO and rGO/NbWO.

A screening process for coating materials is conducted following the acquisition of electrochemical properties using the ACGNet model and the electronic conductivity calculation formula[39]. The initial screening eliminates unsuitable materials containing radioactive, toxic elements, or precious metal elements, ensuring that the remaining candidates offer cost advantages and meet fundamental stability requirements. The dataset after initial screening is presented in Fig. 4b, where each point represents a material with its x, y, and z coordinates, signifying electronic conductivity ($\sigma_e$), ionic conductivity ($\sigma_i$), and oxidation potential ($V_{ox}$), respectively. Further refinement, targeting a $V_{ox}$ above 4 V, $\sigma_e$ above $10^3$ S cm$^{-1}$, and $\sigma_i$ above $10^{-6}$ S cm$^{-1}$, identifies graphene as the most promising coating material with the optimal comprehensive properties, as marked by the red dot in Fig. 4b. Specifically, its electronic conductivity, ionic conductivity, and oxidation potential are $9.2 \times 10^3$ S cm$^{-1}$, $5.61 \times 10^{-6}$ S cm$^{-1}$, and 5.38 V, respectively. Compared to mechanically exfoliated graphene, reduced graphene oxide (rGO) boasts a simpler preparation method and contains a small number of

functional groups, such as carboxyl and hydroxyl (Supplementary Fig. 15), facilitating its integration with organic electrolytes in lithium-ion batteries[40]. Based on the above analysis, rGO was selected as a potential negative electrode coating material and conducted subsequent characterization and electrochemical experiments.

Among various interface control strategies, the rGO, considering its low desolvation energy barrier and high lithium-ion diffusion rate, can guarantee fast Li$^+$ transport and relocation[12,41]. DFT theoretical calculations (Fig. 4c) show that the diffusion energy barrier of lithium ions in rGO (I) is lower than that of migration and diffusion in NbWO tunnel structure (II ~ IV), which indicates that coating rGO in NbWO can effectively increase the overall ionic conductivity of the material, which is conducive to the realization of significantly facilitated diffusion of lithium ions to obtain ultra-high rate performance.

Hence, the NbWO was treated with H$_2$ plasma to improve contact with GO and fabricate the final product of rGO/NbWO[42]. The yellow powder turns gray (Supplementary Fig. 16), indicating the uniform

coating of rGO on the stick-like NbWO particle (Supplementary Fig. 4). EDS elemental mappings show a uniform distribution of C elements on the material surface (Supplementary Fig. 17). As shown in Supplementary Fig. 18, the weight loss from the adsorbed water for NbWO is around 0.7% at 1000 °C, while the weight loss of rGO/NbWO is around 1.2%. This result indicates the weight of the added rGO is about 0.5 wt.%. However, the overall peak position of XPS is shifted after coating the rGO layer, suggesting a change in the chemical state of Nb and W. Furthermore, the C 1$s$ spectra of rGO/NbWO indicate that the material is coated with rGO, and the proportion of C = O in the C 1$s$ peak is small and shows that GO is successfully reduced to rGO (Supplementary Fig. 19)[43]. Meanwhile, compared to NbWO, the Raman spectrum of rGO/NbWO shows distinct D and G peaks, with a D/G peak ratio between 0.988 and 1.053, indicating a higher degree of graphitization on the material's surface, which suggests improved electronic conductivity (Supplementary Fig. 20).

The $I$–$V$ responses of single NbWO and rGO/NbWO crystals, measured via in situ TEM (inset of Fig. 4d), are linear and symmetric in the high-bias regime (5 V). The $I$–$V$ curves exhibit an almost linear relationship (Fig. 4d), allowing the conductance (G) of the crystals to be calculated using G = dI/dV, G was calculated to be $1.69 \times 10^{-3}$ µS for NbWO and $3.45 \times 10^{-2}$ µS for rGO/NbWO, indicating a significant enhancement in electronic conductivity for NbWO due to rGO. Additionally, the powder conductivity was tested using the four-probe method (Supplementary Fig. 21), the electronic conductivity of rGO/NbWO is calculated to be $5.50 \times 10^{-1}$ S m$^{-1}$, while that of the NbWO is only $2.85 \times 10^{-6}$ S m$^{-1}$, indicating that the electronic resistance of rGO/NbWO crystal is small.

As illustrated in Fig. 4e, without any surface modification, lithium ions on the surface of the NbWO crystal can only enter the bulk phase through the tunnel openings, while the shear planes, appearing as the major portion of particle surface area, impede the intercalation of lithium ions, resulting in poor overall lithium-ion diffusion kinetics at these surfaces (the crystal information is provided in Supplementary Data 1-4 and the legends are displayed in description of additional supplementary files). However, coating NbWO with a rGO layer reduces anisotropy at the interface and fosters a uniform Li-rich surface layer, thereby streamlining the diffusion pathway from the shear surface to along the rGO layer, which enhances the transfer of desolvated Li$^+$ ions from the electrolyte to the tunnel pathways[44,45]. As shown in Supplementary Fig. 22, in situ electrochemical impedance spectroscopy (EIS) measurements demonstrate that rGO/NbWO exhibits lower intrinsic impedance (3.83–4.35 Ω at 25 °C) and significantly reduced lithium-ion desolvation impedance at the interface compared to NbWO. Consequently, the enhanced electronic conduction and Li$^+$ transfer in the interfacial region facilitate a more uniform Li-(de) intercalation layer, fostering synchronized electrochemical reactions.

We embarked on a comprehensive electrochemical performance evaluation of the material to delve deeper into the lithium-ion diffusion kinetics of rGO/NbWO. As illustrated in Fig. 5a, solvated lithium ions transition from the electrolyte to the material's surface. The inherent one-dimensional shear structure of NbWO results in uneven diffusion of ions across its surface. However, when the material is encapsulated within an rGO layer, it facilitates the uniform desolvation of Li$^+$ ions, followed by their transport into the rGO shell layer and subsequently into the bulk phase of the material. This phenomenon is primarily evidenced by the impedance changes during the material's charge-discharge cycles. While the material exhibits similar performance during charging and discharging at rates up to 10 C, the performance disparity at higher rates (20–80 C) becomes pronounced. This indicates ionic diffusion predominantly restricts the material's performance at lower rates, whereas, at high rates, the performance is controlled by a synergy of electronic and ionic conductivities. Applying an rGO coating creates a lithium-rich layer, facilitating the rapid transfer of Li$^+$ ions to tunnel openings. This enhancement improves

high-rate performance, achieving 257 mAh g$^{-1}$ at 0.2 C, 168 mAh g$^{-1}$ at 10 C, and 116 mAh g$^{-1}$ at 80 C (Supplementary Fig. 23).

We further compared the rate performance of rGO-coated $Nb_{16}W_5O_{55}$ with other Nb-based negative electrode materials, as shown in Fig. 5b and Supplementary Table 7. Utilizing an interface engineering strategy with a graphene coating, we can swiftly collect desolvated Li$^+$ ions and relocate them to (010) surfaces for rapid intercalation. Consequently, at the micrometer scale, the material exhibits fast charging performance. In the Li‖rGO/NbWO cell, the performance limitations under high rates are not solely attributed to the NbWO material. The ionic conductivity of the electrolytes at different temperatures is presented in Supplementary Fig. 24. It was observed that the logarithm of the lithium-ion conductivity exhibits a strong linear correlation with 1000/T, and the activation energy (E$_a$) of 1 M LiTFSI was calculated as 0.0345 eV by the Arrhenius equation (Eq. 2)[46]. As a comparison, we cycled a Li‖Li symmetric cell at current densities corresponding to the 0.2 C–80 C rates used in Fig. 5a (Supplementary Fig. 25). The overpotential observed in the symmetric cell closely matched the overpotential seen in the electrochemical cycling curves of Supplementary Fig. 23. This indicates that near 25 °C, the limiting factor in fast-charging is largely due to lithium metal plating/stripping or lithium-ion desolvation and transport in the electrolyte, rather than the negative electrode materials.

In addition, the R$_{ohm}$ represents the intrinsic impedance of the material, which is related to the electronic conductivity of the material, and R$_{ct}$ represents the charge transfer impedance of the redox reaction process. As shown in Fig. 5c and Supplementary Table 8, rGO coating can enhance the electronic conductivity of the material and thus significantly reduce the intrinsic impedance of the material. Its lithium-ion migration barrier is much lower than the NbWO, so it can effectively promote the diffusion of lithium ions at the interface and thus reduce the redox reaction impedance. In addition, in order to investigate the electrochemical behaviors during Li-(de)intercalation processes of rGO/NbWO in the 1 M LiTFSI electrolyte, dQ/dV test and CV measurement were applied. The reaction of rGO/NbWO with lithium proceeds in three regions from 2.6 V to 1.0 V with a discharge peak voltage of 1.63 V that can be observed in the dQ/dV curve (Fig. 5d) similar to the CV curves (Fig. 5e). Additionally, the GITT curve of rGO/NbWO (Fig. 5f) shows greater symmetry compared to NbWO (Supplementary Fig. 7), with a lithium-ion diffusion coefficient of up to $3.02 \times 10^{-10}$ cm$^2$ s$^{-1}$ (Supplementary Fig. 26), slightly higher than that of NbWO. This indicates that rGO/NbWO exhibits improved reaction kinetics.

As shown in Fig. 5e, the capacity retention of Li‖rGO/NbWO coin cell is 85.2% (with a decay rate of 0.029% per cycle) for 500 cycles at 1 C, while the capacity retention at 10 C is 92.7% (with a decay rate of 0.015% per cycle), higher than the cycling stability at a lower rate. This anomalous high-rate stability mainly originates from the lattice change of the material during charging and discharging. The behavior of NbWO demonstrates characteristics governed by thermodynamics rather than kinetics. This is evidenced by its response under various electrochemical conditions, where the rate of lithium-ion intercalation and the associated structural changes in NbWO are more significantly influenced by thermodynamic stability and energy minimization than by the rapidity of kinetic processes, hence, the cycling stability at a high rate is higher than that at a low rate.

In order to evaluate the fast charging and discharging application of rGO/NbWO, we illustrated the configuration of a pouch cell, pairing rGO/NbWO with an LFP positive electrode, as depicted in Supplementary Fig. 27. Figure 5g showcases the construction of Ah-like pouch cells tailored for electric vehicle applications, where their fast charging and discharging performances were assessed. In this case, the electrodes were prepared with commercial LiFePO$_4$ powders:Super P:CNT:PVDF = 94:1.5:1.5:3 wt% as the positive electrode (7.3 ± 0.2 mg cm$^{-2}$ for each side and positive electrode loads on two sides) and rGO/NbWO as the

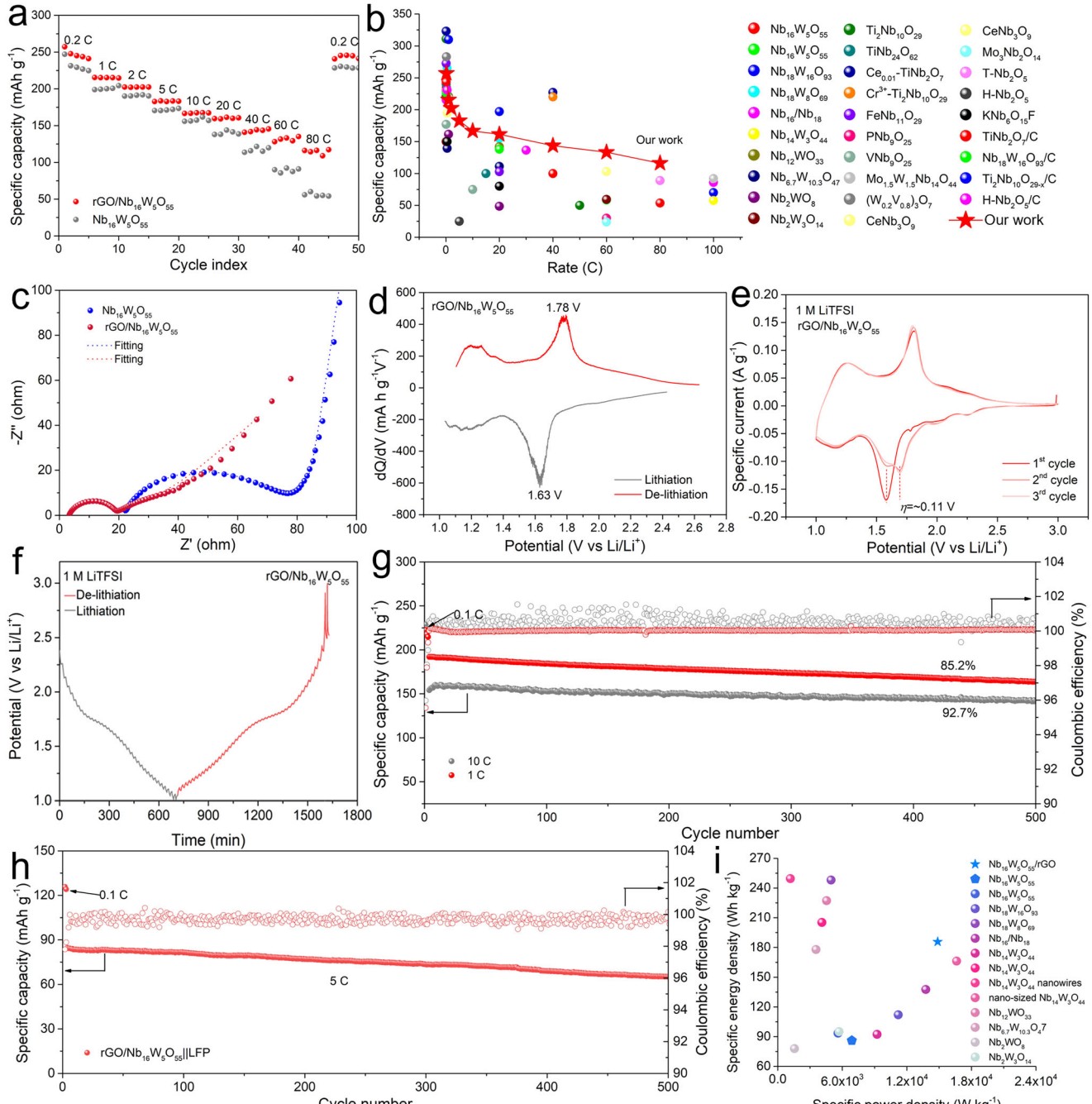

**Fig. 5 | Fast charging performance evaluation. a** Rate performance for rGO/NbWO and NbWO. **b** Comparison of rate performance in different works of literature (the references cited in the Supplementary Table 7). **c** Nyquist plot and fitting data for rGO/NbWO and NbWO. **d** dQ/dV plots of rGO/NbWO in 1 M LiTFSI. **e** The CV curves of rGO/NbWO in 1 M LiTFSI. **f** The GITT curves for rGO/NbWO.

**g** Cycling performance of rGO/NbWO coin cells at 1 C and 10 C. **h** Cycling performance of rGO/NbWO ||LiFePO₄ pouch cell at 5 C. **i** Energy density and power density of NbWO based materials (the references cited in the Supplementary Table 9).

negative electrode in the same way (containing a metal oxide, Super P, CNT, and PVDF with a mass ratio of 94:1.5:1.5:3, 6 ± 0.2 mg cm⁻² for each side and negative electrode loads on two sides). 0.1 Ah pouch cells contain three layers of positive electrodes and four layers of negative electrodes and constructed rGO/NbWO ||LiFePO₄ pouch cells. In addition, the specific discharge capacity of rGO/NbWO ||LiFePO₄ at 5 C was 85 mAh g⁻¹ and remained 96.1% at the first 100 cycles, with a decay rate of 0.04% per cycle. When the pouch cell was fast-charged and discharged for 500 cycles at 5 C, the capacity could still be maintained at around 77.0% with a decay rate of 0.16% per cycle (Fig. 5g). However, the current performance in practical applications is not ideal, primarily due

to several limiting factors. As rGO/NbWO is a fast-charging negative electrode material, most performance evaluations have been conducted with low loading levels (1–3 mg cm⁻²). When the loading increases, its rate performance deteriorates significantly. For instance, the rGO/NbWO with an areal capacity of around 0.44 mAh cm⁻² achieves 257 mAh g⁻¹ at 0.2 C, when the loading is increased to 15 mg cm⁻² with the areal capacity reaching 2.88 mAh cm⁻², the impedance rises to 198 Ω, and the capacity is 208 mAh g⁻¹ at 0.2 C and drops to 24 mAh g⁻¹ at 40 C (Supplementary Fig. 28). Additionally, the commercial LFP shows a significant decline in rate performance under high current conditions, which results in the relatively low capacity of the rGO/NbWO ||LiFePO₄ pouch cell

(Supplementary Fig. 29). Despite these challenges, Fig. 5h and Supplementary Table 9 show that $rGO/Nb_{16}W_5O_{55}$ electrode, with a tap density of $1.61\,g\,cm^{-3}$, can achieve an impressive energy density of $186\,Wh\,kg^{-1}$ at a high power density of $14,860\,W\,kg^{-1}$, and $406\,Wh\,kg^{-1}$ at $81\,W\,kg^{-1}$, surpassing most of NbWO based materials. Furthermore, using the BatPac model developed by Argonne National Laboratory[47], we simulated a 100 Ah $rGO/Nb_{16}W_5O_{55}$||LFP cell with positive electrode and negative electrode areal capacities reaching $3.8\,mAh\,cm^{-2}$. This simulation demonstrated a volumetric energy density of $251\,Wh\,L^{-1}$ and a gravimetric energy density of $123\,Wh\,kg^{-1}$.

## Discussion

In conclusion, we demonstrate here that the fast-charging capability of NbWO is primarily due to a rate-dependent lattice relaxation process. The NbWO structure exhibits a high sensitivity to charging rates, with lithium ions randomly occupying positions at higher rates, thereby mitigating the substantial lattice distortions characteristic of the NbWO structure and enhancing fast $Li^+$ ions intercalation. A notable observation was that lattice distortions were accentuated at lower rates, such as 0.1 C, which compromises structural stability. Furthermore, [010]-preferred $Li^+$ transport in $Nb_{16}W_5O_{55}$ crystals was identified as a critical bottleneck for achieving fast charging. To address these challenges, we employed a machine learning-assisted interface engineering strategy to create a lithium-rich layer, facilitating the rapid transfer of $Li^+$ ions to (010)-faceted entrance, alleviating the spatial and temporal anisotropy of $Li^+$ (de)-intercalation due to local inhomogeneities during fast charging and discharging cycles. As a result, the as-engineered NbWO (rGO/NbWO) exhibits a capacity of $257\,mAh\,g^{-1}$ at 0.2 C and $116\,mAh\,g^{-1}$ at 80 C, which maintains a capacity of 92.7% for 500 cycles at a rate of 10 C. In addition, the rGO/NbWO||$LiFePO_4$ pouch cell cycled at 5 C and remained 96.1% at the first 100 cycles and 77.0% capacity after 500 cycles at a large rate of 5 C. We anticipate that these direct, atomic-scale findings will deepen understanding of the fast-charging mechanisms of $Nb_{16}W_5O_{55}$-based batteries.

## Methods

### Synthesis of $Nb_{16}W_5O_{55}$

NbC powder (from Aladdin) was annealed in air at 950 °C for 10 h to produce H-type $Nb_2O_5$ (H-$Nb_2O_5$) powders. These powders were then sifted through a 400 mesh screen. H-$Nb_2O_5$ and $WO_3$ (from Aladdin) were weighed to achieve an 8:5 stoichiometric ratio with a precision of 0.01 g. The powders were thoroughly blended using ball milling, followed by manual grinding, before being transferred to a crucible. Subsequently, the powder mixture was heated to 700 °C in an air atmosphere at a heating rate of 5 °C/min, maintained at this temperature for 12 h, and then further heated to 1200 °C for an additional 12 h.

### Synthesis of $rGO/Nb_{16}W_5O_{55}$

The GO solution was prepared by dissolving 0.2 g of GO (acquired from XFNano) in 100 ml of deionized water, followed by sonication using an ultrasonic cell disintegrator in an ice bath for 300 min, with a cycle of 5 s on and 5 s off, at approximately 70% power. This process resulted in a 2 mg/ml GO solution without any precipitation. Subsequently, 1 g of $Nb_{16}W_5O_{55}$, pretreated with $H_2$-plasma for 5 min, was added to the GO solution and stirred continuously for 12 h. The mixture was then centrifuged, followed by drying in an oven at 80 °C for 12 h. The resultant $GO/Nb_{16}W_5O_{55}$ powders were annealed at 300 °C for 15 min in an air atmosphere, producing $rGO/Nb_{16}W_5O_{55}$.

### Materials characterizations

SEM images were obtained using a TESCAN MIRA4 scanning electron microscope. STEM analyses were conducted on an aberration-corrected STEM, specifically the JEM-ARM300F microscope (manufactured by JEOL, Ltd, Japan), operating at 300 kV. XPS measurements were carried out using a Thermo Scientific K-Alpha+ instrument.

Particle size analysis was performed using a Malvern Mastersizer 2000, with each sample being measured thrice to obtain an average value.

### Electrochemical characterization

Electrochemical evaluations were performed using 2032 coin cells composed of an electrode ($2-15\,mg\,cm^{-2}$, 12 mm in diameter and coat one side), a Whatman glass microfiber separator (the Whatman GF/D glass microfiber separator has a thickness of 0.22 mm, porosity of 80–85%, average pore size of $1.0\,\mu m$, and is supplied by Whatman (Cytiva)), a 15.8 mm-diameter lithium metal disk (the lithium is from Ganfeng Lithium (99.9%), 15.8 mm diameter, 0.6 mm thick, stored in a glovebox at 25 °C), two stainless-steel metallic gaskets, and a stainless-steel conical spring. The electrode formulation comprised metal oxide, conductive agent, and binder in an 80:15:5 mass ratio applied to copper foil. The active material, conductive agent (Super P), and PVDF (Solef 5130) were manually mixed in an agate mortar and pestle. This mixture was then dispersed in 1.5 mL of N-methyl-2-pyrrolidone (NMP) with 4% weight of homogeneously dispersed carbon nanotubes (CNTs). After 6 h of stirring, the resultant slurry was spread onto copper foil and dried at 120 °C for 12 h under vacuum conditions. The electrolyte used was 1.0 M lithium bis(trifluoromethanesulfonyl)imide (LiTFSI, Canrd, battery grade) in a 1:1:1 volume ratio of dimethyl carbonate, ethyl methyl carbonate, and ethylene carbonate (DMC:EMC:EC, Aladdin, battery grade), the electrolyte stored in a glass box in glovebox with Ar at 25 °C and used in 24 h. The electrolyte is added dropwise, with 60 μL per coin cell, using a micropipette with a 100 μL volume capacity and disposable tips. Electrochemical testing was carried out in a climate-controlled environment at $293 \pm 2\,K$ using EC-Laboratory and LAND software. The half-cells operated within a voltage range of 3.0 V to 1.0 V with respect to Li/$Li^+$, and the charge rate (C-rate) was standardized to $1\,C = 170\,mA\,g^{-1}$. The specific capacity is calculated based on the mass of the $Nb_{16}W_5O_{55}$ active material. The Coulombic efficiency (CE) is calculated as the ratio of discharge capacity divided by the charge capacity in the preceding charge cycle, independently of whether a cell/battery is assembled in a charged or discharged state. The galvanostatic cycling tests were conducted using a LAND battery testing system, and the data were directly processed and plotted using Origin 9.0.

To further indicate the commercial practicability of $rGO/Nb_{16}W_5O_{55}$, the cycle with high-ratio performance was measured by the full battery, and the $LiFePO_4$ acted as the positive electrode. For the positive electrode, the rate of $LiFePO_4$ powders:Super P:CNT:PVDF is 94:1.5:1.5:3 w% and coating on the Al foil. In the initial, the 3.0 g binder agent was dissolved in 45 g NMP and solved by mechanical stirring in a stainless-steel tank. Following the 1.5 g conduct agents, and 37.5 g CNT homodisperse liquid (the 4 w% of CNT) was added into the mixture fluid to homodisperse at a high-speed stirring. At last, the 94 g $LiFePO_4$ active materials were added and kept speed stirring for 6 h. All experiment processes were carried out in a low-humidity environment, and the total solid content of polarity pulp hold at 55%. The mass loading of the $LiFePO_4$ was $7.3 \pm 0.2\,mg\,cm^{-2}$ coating on the Al foil. Meanwhile, the areal density of the $rGO/Nb_{16}W_5O_{55}$ electrode, containing a metal oxide, Super P, CNT, and PVDF with a mass ratio of 94:1.5:1.5:3, was controlled in $6 \pm 0.2\,mg\,cm^{-2}$ coating on the Cu foil through the calculations based on the charge balance.

In this work, a pouch cell was assembled using three positive electrodes and four negative electrodes, with the $LiFePO_4$ electrode and $rGO/Nb_{16}W_5O_{55}$ electrode sliced to 35 mm × 52 mm and 39 mm × 56 mm, respectively. The electrodes were laminated in a zigzag pattern with a PP separator. Due to the small cell volume, no additional gas release bag was required. After removing the moisture from the cell, the 2.0 g 1 M LiTFSI electrolyte was injected according to an appropriate E/C ratio (Electrolyte/Capacity = $4.0\,g\,Ah^{-1}$). To further analyze the cycle performance of the materials, the pouch cell was first activated at a 0.1 C rate for 3 cycles, followed by testing at 5 C rate in a voltage window of 1.0–2.5 V with charge/discharge and tested at

$293 \pm 2$ K in an air-conditioned environment. The specific capacity for pouch cell is calculated based on the mass of the $LiFePO_4$ active material. The testing was conducted under a pressure of 0.3 MPa, applied using a limiting fixture.

The in situ EIS performances of NbWO-based electrodes were investigated by an Autolab 302 N at $293 \pm 2$ K in an air-conditioned environment with a voltage range of 1 V to 3 V and the frequency range from 0.01 Hz to 100,000 Hz. A 5 mV voltage perturbation was applied, and the cell was left to rest at open-circuit potential for 12 h before measurements. A total of 71 data points were collected, spaced logarithmically across the frequency range.

The energy density and power density for $rGO/Nb_{16}W_5O_{55}$ materials are calculated by the mass of $rGO/Nb_{16}W_5O_{55}$. Additionally, using the BatPac model developed by Argonne National Laboratory, we simulated a 100 Ah $rGO/Nb_{16}W_5O_{55}||LFP$ cell, where the areal capacities of the positive and negative electrodes reach 3.8 mAh cm$^{-2}$. The volumetric and gravimetric energy densities are calculated based on the total volume of the pouch cell.

### Computational methods

All spin-polarized calculations were performed with the help of the CASTEP package in Material Studio. The PBE exchange-correlation method was selected to describe the electronic interactions in the systems. DFT-D dispersion correction was carried out to accurately simulate the van der Waals and non-bond interactions. A plane wave basis set with an energy cutoff of 520 eV was employed for self-consistent calculations and structural optimization. The relaxation of slabs was performed until the total energy per atom converged to less than $1.0 \times 10^{-5}$ eV, and atomic forces were reduced below $-0.05$ eV Å$^{-1}$. The K-point grid was sampled using the Monkhorst-Pack method with dimensions of $1 \times 1 \times 1$[48]. Hubbard U with Nb $4d$ (4.00 eV) and W $5d$ (4.00 eV) were set to minimize the electronic self-interaction error in strongly correlated orbital.

### Machine learning-assisted interface engineering strategy

Coating materials generally need to prioritize meeting relevant requirements, such as high electronic and ionic conductivity, as well as a high oxidation potential to maintain electrochemical stability. Therefore, we first used the ACGNet model to predict the ionic conductivity $\sigma_i$ and oxidation potential $V_{ox}$ (versus Li$^+$/Li) of each material[39]. ACGNet is a deep learning model based on graph neural networks that can predict the corresponding properties of input material crystal structures. The training data of ACGNet comes from the dataset provided by Austin D. Sendek et al. [49]. The dataset includes the crystal structures of 40 crystalline materials and their bulk ionic conductivities at 25 °C, as well as the crystal structures and oxidation potentials of 216 other materials. Subsequently, the electronic conductivity of each material was approximated using a conductivity calculation formula based on carrier mobility and volume concentration[50]. The electronic conductivity for an intrinsic crystalline semiconductor can be approximated as:

$$\sigma_e = (\mu_e + \mu_h) q \sqrt{N_c N_v} e^{-E_{gap}/2kT} \qquad (1)$$

Where $\mu_e$ and $\mu_h$ denote the electron and hole mobilities, $q$ is the charge of an electron, $N_c$ and $N_v$ represent the state densities in the conduction and valence bands, respectively. The standard values for crystalline silicon ($N_c = 2.89 \times 10^{19}$ cm$^{-3}$, $N_v = 3.13 \times 10^{19}$ cm$^{-3}$, $\mu_e = 1430$ cm$^2$V$^{-1}$ s$^{-1}$, $\mu_h = 480$ cm$^2$V$^{-1}$ s$^{-1}$, T = 300 K) are employed to derive the electronic conductivity of materials at 25 °C. Although the parameters in Eq. (1) vary across materials, the general trend of a larger bandgap leading to lower electronic conductivity remains consistent. Consequently, this formula, in conjunction with the bandgap of each material, is utilized to crudely estimate the electronic conductivity of each material.

### Electrolyte ionic conductivity measurement

The temperature dependence of lithium-ion conductivity is characterized by the Arrhenius equation, expressed as[46]:

$$\sigma = A\exp^{\frac{-E_a}{kT}} \qquad (2)$$

where A is the pre-exponential factor, $E_a$ denotes the activation energy for lithium-ion conductivity, and $k$ is the Boltzmann constant.

## Data availability

All data generated or analyzed during this study are included in this published article (and its supplementary information files). All the text, images, and videos in this manuscript are written or experimentally obtained by authors and are not related to large language models or AI tools. Hence, there is no prompt. Only the property prediction model ACGNet belongs to traditional machine learning algorithms and is cited from the work provided by Wuxin Sha et al. (https://doi.org/10.1016/j.electacta.2023.143459). ACGNet is a traditional graph convolutional neural network model without the prompt words. ACGNet is executed in Python language (version 3.10.2), with main computation packages of NumPy (version 1.22.3) and PyTorch (version 1.11.0). The data generated in this study are provided in the Source Data file. Source data are provided with this paper.

## Code availability

The database used for screening is a complete copy of the Materials Project database as of 10/18/2018, which contains 83989 entries (https://materialsproject.org). The database used for model training comes from the dataset provided by Sendek et al. (https://doi.org/10.1039/C6EE02697D).

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

## Acknowledgements

We acknowledge the National Key R&D Program of China (2021YFA1202300 and 2024YFA1211900, received by Y. Y.). This work was financially supported by WenZhou-2024R3001 (received by Y. Y.). Dr. Anmin Nie appreciates the support of the National Natural Science Foundation of China (52090022 and 52288102, received by A. N.). This work gratefully acknowledges support from the U. S. Department of Energy (DOE), Office of Energy Efficiency and Renewable Energy, Vehicle Technologies Office. Argonne National Laboratory is operated for DOE Office of Science by UChicago Argonne, LLC, under contract number DE-AC02-06CH11357 (received by T. L.).

## Author contributions

Conceptualization: Y.G., Y.Y., K.H., T.L., and A.N. Methodology: Y.G., Y.Y., T.L., A.N., and K.H. Formal analysis: Y.G., Y.Y., and A.N. Investigation: Y.G., P.L., W.H., J.Y., X.L., C.G., W.S., X.Z., X.T., and Q.R. Resources: Y.G., Y.Y., S.W., T.L., A.N., and K.H. Supervision: Y.Y., T.L., and A.N. Project administration: Y.Y., T.L., and A.N. Funding acquisition: Y.Y., T.L., and A.N. Writing—original draft: Y.G. Writing—review and editing: Y.G., C.G., X.L., X.Z., Y.Y., K.A., and A.N.

## Competing interests

The authors declare no competing interests.
