## [Transparent Peer Review file · Nature Communications]

Unleashing Faster Charging in NbWO-based Batteries

Corresponding Author: Dr Tongchao Liu

Version 0:

Reviewer comments:

Reviewer #1

(Remarks to the Author)

The authors present a new approach to understanding the mechanisms of fast charging in NbWO batteries and the limitations on current charging times. Although the material is well known, the authors were able to demonstrate that high-speed Li intercalation is linked to a network relaxation process. They were able to demonstrate that high-speed charging was linked to a process of relaxation of the material lattice combined with a directional effect of lithium diffusion and transport along a preferred crystallographic axis. This directional effect has been identified as a limiting factor in fast charging processes.

To overcome these problems of material anisotropy, the authors applied a coating of rGO, which not only increased electrical impedance but also lowered the barrier to lithium ion diffusion. All these results are perfectly supported by a solid experimental aspect that combines a set of impressive characterization techniques, the quality of the results of which support the authors' conclusions.

Finally, to demonstrate the efficacy of their new material and test their hypothesis and conclusions based on experimental results, the authors fabricated a pouch cell battery with the LFP/rGO||NbWO system and demonstrated the battery's fast-charging reliability during the first 100 cycles and a 23% loss after 500 cycles.

By understanding the mechanisms of high-speed Li⁺ intercalation and the physical effects associated with its limitation, this work has enabled them to propose a new material design that significantly enhances its ultra-fast charging properties. I hereby recommend this work for publication in the journal Nature Communications.

Reviewer #2

(Remarks to the Author)

This paper reports Nb₁₆W₅O₅₅ for a high-power LIB anode material. The followings are my technical concerns.

1. First of all, the manuscript is not well organized in my opinion. The authors start explanation of the fast charging mechanism of Nb₁₆W₅O₅₅ using STEM (Figure 2) before explaining its electrochemical performance (Figure 5). Thus, during the first reading, I could not understand the logic. I suppose thorough rewriting/restructuring is needed.
2. The authors claim that Li sites change depending on the intercalation rate. However, if Li sites change, Li intercalation voltage must be altered largely. I propose to compare DFT-calculated voltage profiles with experimental data with different C rates.
3. In situ TEM was conducted in the all solid state cell, which is largely different from the environment of a cell with a liquid electrolyte. The electrochemistry of Nb₁₆W₅O₅₅ should differ because it depends on the C rate, as claimed by the authors. I do not recommend to include this misleading data in this excellent work.
4. rGO is a well-known material to improve the electrode performance. The authors claim that data science proposed rGO, but no quantitative analysis/discussion is provided in the main text, making this approach unconvincing. This section needs further improvement, for example, by predicting electrode performance improvement effect quantitatively using the features of physical properties (electronic conductivity, ionic conductivity, etc.).
5. Although the authors discuss an interesting solid-state electrochemistry of Nb₁₆W₅O₅₅ in the first half of the manuscript, which I believe is worthy of publication in Nat. Commun., the other half focus on the development of rGO/Nb₁₅W₅O₅₅

composite and full cell, which were conducted routinely. I can find many similar papers in many journals, which is not relevant with either basic science or practical battery technology. Therefore, I strongly recommend to thoroughly rewrite the manuscript by focusing on the rate-dependent electrochemistry of Nb₁₆W₅O₅₅.

Reviewer #3

(Remarks to the Author)

The paper has a very general title which piqued my interest. However, the content turned out to be very, very specific and related to detailed aspects of a single material. My first suggestion would be to change the title, otherwise readers may also be confused. Being a more general materials scientist/electrochemist, I had some difficulty in assessing the paper due to the wealth of detailed information. I suggest that some dedicated experts should also review this paper. In terms of analytical tools, the paper seems to be at a very high level. However, I have some general comments and questions. They largely address the discussion of the results (for the moment I do not see the need for more experiments. If the points are properly addressed, I think the paper can be a very valuable contribution to the battery research community.

Authors identify rGO as most promising coating substance. A better description of rGO should be provided. rGO is generally a very ill-defined material so it remains unclear why it comes out as first choice. What is the actual reason? How does the applied rGO compares to the structure of the rGO from the materials database?

It is not clearly described enough, why rGO is effective in improving the anisotropy of the particles. The explanation on page 16/17 sounds very general and seems a hypothesis only. But is there any evidence?

Authors state values for the specific capacity with an extra digit, e.g. 257.2 mAh/g. But is this reproducible? Authors may round number to e.g. 257 mAh/g.

Authors correctly mention in the introduction that nanosizing of Si anodes (or Si/carbon anodes) leads to a penalty in volumetric energy density. Authors should comment on the volumetric energy density (and gravimetric energy density) of their material and cells.

Generally, fast charging can be obtained by decreasing the areal loading of the electrode. Authors should comment on that and discuss their results. What is the areal capacity compared to conventional high rate batteries (mAh/cm²)?

Authors motivate their study clearly by application (military and space applications). So I miss a balanced discussion on energy density and fast charging. For example also a plot, where existing high rate batteries are compared to the cell the authors make. Maybe a Ragone plot? From the provided information, I could not yet see how relevant the findings are in practice. The electrode of the authors also contains 15 wt% carbon additive and 5 wt% binder. This is OK for research purposes but if authors claim relevance for application it should be compared to composition of state-of-art anodes. I also wonder under which conditions the anode is rate limiting. Is the LFP cathode always better than the anode the authors made? Or is the prepared anode faster than the LFP cathode? Authors may also mention that rate performance is also highly temperature dependent. Maybe it is too difficult to clarify all these questions but authors should at least mention and discuss these points.

Some statement should be rewritten for better clarity, e.g. "Graphite and silicon as LIB anode can store a large amount of Li⁺ ions within the potential range close to the Li⁺/Li redox couple; in practice, however, graphite stores Li⁺ at a low potential close to the Li plating reaction,...". The way the statements are connected make not too much sense.

Another example: "when the Li⁺ ions that have already been desolvated from the electrolyte molecules get well prepared for their entry into the lattice.."

Last sentence of paper: "...pouch cells can charge mobile phones, and this development is expected to become the mainstream trend of future shared power". I suggest to remove such as statement. I do not understand its scientific value.

Version 1:

Reviewer comments:

Reviewer #1

(Remarks to the Author)

The manuscript can be accepted as it has been clarified with all answers to review comments

Reviewer #2

(Remarks to the Author)

The manuscript was revised appropriately, so it's now suitable for publication in Nat. Commun.

Reviewer #3

(Remarks to the Author)

The manuscript clearly improved - again with a wealth of data. While this is useful and greatly acknowledged, the authors should simply discuss some of the data clearer in order to make the right claims. After these corrections, the paper could fulfill the standards of Nat Comm.

“Supplementary Fig. 15. The FTIR spectrum of rGO/Nb16W5O55.” Needs a more critical discussion as the signal/noise is very weak. Also: What happens below 1250 cm^{-1} ? Figure S20 at least contains a comparison.

Figures S21: Details of the four probe method need to be mentioned so that other people can reproduce.

Figure S22 needs to be replotted. There is hardly anything seen. Please provide information on the equivalent circuit and the quality of the fit. Indicate relevant frequency values. What was the frequency range of the experiment? Lot of things are unclear, experimental part should be more details on how the measurements were made. EIS data needs to be plotted in a x-y diagram with equal scaling of the axis.

I acknowledge that the authors used a large database to identify graphene as best material. Reduced graphene oxide, however, has not the same properties. Authors now state in the rebuttal “rGO also retains the electrochemical characteristics of conventional graphene stemming from its crystalline structure and size effects. Based on the above analysis, rGO was selected as a potential anode coating material ...”. This statement should be clearly reconsidered. For example, the oxidative stability of graphene will be very, very different from graphene oxide.

Figure 4: Typo: Algorithm

Figure 4c: Should be improved. Y-axis has no numbers. Indicate the migration paths. I can not understand much from the illustration so far.

Figure 4d: Letter size in TEM images needs to be increased. It is not possible to read.

Energy density calculation: This is now clearer but it should be stated that the values refer to the active materials only. In addition, authors used BatPac, which is good but information on what parameters were used is missing, e.g. what loading was assumed (0.44 mAh/cm^2 ?). Maybe state the values in the supp.info?

For the last sentence on the energy density: “...suggesting promising potential for practical applications”, authors should compare to LTO/LFP data. If not, I would not make the claim but simply focus on the scientific discoveries of the paper. The low loading of 0.44 mAh/cm^2 seems not close to application.

Figure 5: Scale coulombic efficiency more meaningful, e.g. from 80 – 110 %. When plotting it from 0-120% it always looks constant.

Version 2:

Reviewer comments:

Reviewer #3

(Remarks to the Author)

Authors expanded on their discussion. The manuscript is now publishable.

Point-by-point responses to reviewers' comments

(Blue type: Reviewer's comments; Black italic: Our response; Red type: Our revised)

Reviewer #1:

The authors present a new approach to understanding the mechanisms of fast charging in NbWO batteries and the limitations on current charging times. Although the material is well known, the authors were able to demonstrate that high-speed Li intercalation is linked to a network relaxation process. They were able to demonstrate that high-speed charging was linked to a process of relaxation of the material lattice combined with a directional effect of lithium diffusion and transport along a preferred crystallographic axis. This directional effect has been identified as a limiting factor in fast charging processes.

To overcome these problems of material anisotropy, the authors applied a coating of rGO, which not only increased electrical impedance but also lowered the barrier to lithium-ion diffusion. All these results are perfectly supported by a solid experimental aspect that combines a set of impressive characterization techniques, the quality of the results of which support the authors' conclusions.

Finally, to demonstrate the efficacy of their new material and test their hypothesis and conclusions based on experimental results, the authors fabricated a pouch cell battery with the LFP/rGO||NbWO system and demonstrated the battery's fast-charging reliability during the first 100 cycles and a 23% loss after 500 cycles.

By understanding the mechanisms of high-speed Li⁺ intercalation and the physical effects associated with its limitation, this work has enabled them to propose a new material design that significantly enhances its ultra-fast charging properties.

I hereby recommend this work for publication in the journal Nature Communications.

Response to Reviewer 1: Thank you for the careful review on our work, particularly the agreement with its significance in revealing the fast-charging mechanisms and kinetics limitations in NbWO anode. We appreciate your fair evaluation of our work.

Reviewer #2:

This paper reports Nb₁₆W₅O₅₅ for a high-power LIB anode material. The followings are my technical concerns.

Response to Reviewer 2: We greatly appreciate the reviewer's careful review of our work. We have carefully addressed the reviewer's comments in the revised manuscript. Please find our detailed responses to your comments below.

1. First of all, the manuscript is not well organized in my opinion. The authors start the explanation of the fast charging mechanism of Nb₁₆W₅O₅₅ using STEM (Figure 2) before explaining its electrochemical performance (Figure 5). Thus, during the first reading, I could not understand the logic. I suppose thorough rewriting/restructuring is needed.

Response to comment-1: Thank you for your valuable comment. We appreciate your comments regarding the organization of the manuscript, and we understand how the flow of our narrative may have caused some confusion. We would like to clarify the logic behind our structure/figure arrangement and explain how it was carefully designed to convey the intrinsic fast-charging mechanism of the Nb₁₆W₅O₅₅ material and how to make fast-charging batteries charging faster.

In brief, the logic of our manuscript and figure arrangement is clarified as following:

Fig. 1 is to show the basic atomic structure of the NbWO anode for fine understanding;

Fig. 2 is to explain why this material can be fast-charged, as previously tested in terms of the decent high-rate performance in literature;

Fig. 3 is to explain why this material is still insufficient for even faster charging (within one minute), which is herein revealed by in situ TEM to be due to the anisotropic limitation;

Fig. 4 is to demonstrate how an efficient surface strategy is screened out and applied to the material so as to address the anisotropic issue;

Fig. 5 is therefore arranged at the last to show how the aboved-proposed surface strategy efficiently improve the ultra-fast rate performance.

For detailed clarifications of our manuscript/figure logic, please see following:

In our manuscript, we started with Fig. 1, **which characterizes the morphology and structure of Nb₁₆W₅O₅₅**. We showed that Nb₁₆W₅O₅₅ crystals grow as elongated rods along the [010] axis, with distinct structural features along different crystallographic axes. Specifically, the crystal structure imaged along the [010] zone

a
x
i

s In Fig. 2, we **found that the fast-charging nature of such material originates from an interesting rate-dependent lattice relaxation process associated with the Jahn-Teller effect**. To investigate this, we conducted ex-situ ABF STEM imaging at various lithiated states at 0.1 C, 20 C and 80 C. Our findings revealed how the position of Li⁺ within the lattice and the response of the Nb₁₆W₅O₅₅ structure depend on the lithiation state. Specifically, for the pristine host, the random occupation of Nb and W sites results in longer Nb-O bonds compared to W-O bonds, leading to a non-linear arrangement of oxygen atoms. As a result, the ABF image of the pristine crystal appears disordered. However, upon partial lithiation at a low rate (0.1 C), the initially highly distorted host crystal undergoes lattice relaxation due to the strengthening of Li-O bond interaction, leads to the recovery of the crystal ordering. Upon further lithiation to a larger amount, the crystal exhibits thermodynamic instability with subsequent lattice collapse. For lithiation at higher rates (20 C and 80 C), the fast-charging capability of NbWO is found to be thermodynamically controlled rather than kinetically driven, which could be achieved by the initial fast Li⁺ intercalation step with the removal of crystal distortions to further facilitate rapid intercalation of Li⁺ ions through regular tunnel sites.

n **While these intrinsic properties enable fast charging, they also impose**

t
o

significant limitations under even higher rate conditions. To further investigate these limitations, we conducted in situ TEM experiments in Fig. 3, revealing that the anisotropic lattice change upon lithiation confines Li^+ intercalation primarily to the [010] axis. This directional, [010]-preferred lithiation mechanism acts as a bottleneck for higher-rate operation, since it prevents the entry of desolvated Li^+ into the host through any non-(010) surfaces.

To address this issue, in Fig. 4 we developed a **machine learning-assisted surface strategy incorporating rGO, with an aim of efficiently collecting and redirecting the desolvated Li^+ to the (010)-terminated surfaces for rapid intercalation into the host instead of being re-solvated into the electrolyte.**

To test the effectiveness of the strategy proposed in Fig. 4, we **compared the performance of the material before and after the surface modification strategy, in Figure 5, and found that surface engineering indeed significantly improved the material's electrochemical performance.** We achieved a capacity of $\approx 116 \text{ mAh g}^{-1}$ (68.5% of the theoretical capacity) at an 80 C rate when coupled with a Li anode.

We recognize that the writing of our manuscript may differ from the conventional narrative style featuring structure-performance-mechanism discussion sequence, which is typically seen in materials research. In our manuscript, we placed a strong emphasis on elucidating the intrinsic mechanisms of fast charging before delving into the electrochemical performance, which might have led to some misinterpretation. To address this, we have improve the clarification of our logic by **adding a more detailed discussion of the electrochemical performance of the unmodified NbWO material**, tightly integrating it with the explanation of the fast-charging mechanism. The specific changes (On manuscript Page 7) and supporting information (on supporting information Page 8) include:

“Fast charging mechanism of $\text{Nb}_{16}\text{W}_5\text{O}_{55}$ crystal

The reaction of $\text{Nb}_{16}\text{W}_5\text{O}_{55}$ with Li proceeds in three distinct regions from 3.0 V to 1.0 V in the CV curve (Supplementary Fig. 6a), these regions are characterized by the slopes of the voltage profile (Supplementary Fig. 6b). The curve exhibits a

noticeable peak occurring at 1.64 V during the lithiation process, primarily corresponding to the reduction of Nb^{5+} . Notably, NbWO demonstrates a high lithium-ion diffusion coefficient at 2.1 V and 1.7 V during the intercalation process of Li^+ , as determined by the galvanostatic intermittent titration technique (GITT, illustrated in Supplementary Fig. 7). In addition, at a rate of 0.2 C, approximately 1.3 Li^+ per transition metal (Li^+/TM) can be reversibly intercalated, resulting in a capacity of about 227 mAh g^{-1} (Supplementary Fig. 6c). When the rate is increased to 5 C, $\text{Nb}_{16}\text{W}_5\text{O}_{55}$ maintains a capacity of 170 mAh g^{-1} . At this rate, the reaction plateau remains evident; however, when the rate is further increased to 20 C and beyond, which corresponds to a three-minute discharge, the slope of the curve significantly increases. This obvious change is likely related to alterations in the crystal structure of the material under different rates. Consequently, we conducted a detailed investigation into the crystal structure of the material at various rates.

Supplementary Fig. 6. The a) CV curves, b) charging/discharging curves, and c) rate performance of $\text{Nb}_{16}\text{W}_5\text{O}_{55}$ in 1 M LiTFSI.

Supplementary Fig.7. The GITT curves and the chemical diffusion coefficients of Li^+ (D_{Li^+}) plots for $\text{Nb}_{16}\text{W}_5\text{O}_{55}$. ”

2. The authors claim that Li sites change depending on the intercalation rate. However, if Li sites change, Li intercalation voltage must be altered largely. I propose to compare DFT-calculated voltage profiles with experimental data with different C rates.

Response to comment-2. Thank you for your constructive comment regarding the relationship between Li site changes and the intercalation voltage. We agree that if Li sites change with the intercalation rate, the Li intercalation voltage should be altered accordingly. Following your suggestion, we compared the DFT-calculated voltage profiles with experimental data at different C rates. At a low rate of 0.1 C, the diffusion time for Li ions is sufficiently long, allowing them to arrange themselves according to the principle of minimum energy. As a result, the DFT-calculated voltage curve aligns well with the experimental curve. However, at a high rate of 20 C, the shorter diffusion time leads to random Li ion arrangements. This randomness is substantial, making it difficult to accurately calculate the voltage. We have revised the manuscript to reflect this point on manuscript Page 11, with the following addition:

“Additionally, DFT calculations were used to simulate the voltage curves of NbWO at homogeneous and heterogeneous intercalation models. The results show that at a homogeneous intercalation model (corresponding to low discharging rates), each NbWO cavity is occupied by one Li^+ following the minimum energy principle, resulting in a distinct discharge plateau in the voltage curve, consistent well with the charge-discharge curves. However, in the heterogeneous intercalation model (corresponding to high discharging rates), due to the randomness of Li^+ occupancy, the calculated voltage fluctuates significantly and decays rapidly (Supplementary Fig. 13).”

3. In situ TEM was conducted in the all solid-state cells, which is largely different from the environment of a cell with a liquid electrolyte. The electrochemistry of $\text{Nb}_{16}\text{W}_5\text{O}_{55}$ should differ because it depends on the C rate, as claimed by the authors. I do not recommend including this misleading data in this excellent work.

Response to comment-3: Thank you for your insightful comment. Upon careful review, we realize that there may have been some confusion due to unclear explanations in the original version of the manuscript. We would like to clarify that the *in situ* solid TEM

was not intended to study the surface-associated battery behaviors, such as solid-liquid or solid-solid interface reactions. Instead, our *in situ* solid TEM was to investigate the lithiation behavior within the bulk of the material and to understand the otherwise hidden anisotropy. In fact, *in situ* solid TEM has been widely used to study various materials, including alloy-type anodes (*Adv. Funct. Mater.* 2024, 2410840), conversion-type anodes (*Adv. Mater.*, 2008, 20:4269), and the intercalation-based anode materials in our research. For example, *in situ* solid TEM has been applied to investigate lithium-ion transport within the bulk of Si anodes. In particular, the anisotropic expansion behavior of silicon upon lithiation observed via *in situ* solid TEM (*Nano Lett.* 2011, 11, 8, 3312–3318) has also been observed via *in situ* TEM using electrochemical liquid cell with real liquid electrolyte (*Nano Lett.* 2013, 13, 6106–6112) and *ex situ* studies of the practically cycled Si material (*Nano Lett.* 2011, 11, 3034–3039, *Adv. Funct. Mater.* 2011, 21, 2412–2422), demonstrates the reliability of *in situ* solid TEM to investigate the lithiation mechanisms within the bulk electrode materials. Based on this, we believe that our *in situ* solid TEM investigation of bulk ion transport and anisotropy in our work is highly relevant and reliable to reflect the true mechanisms within a cell of liquid electrolyte.

In addition, our primary goal was to determine the factors limiting lithiation, specifically the role of anisotropy, which we believe is the key limiting factor. Hence, we included *in situ* TEM characterization of the lithiation behaviors along different crystallographic axes and discovered that Li^+ ions can only undergo rapid intercalation/extraction through the [010] direction. However, in the stick-like NbWO, which is also elongated along [010], the exposed surface of the particle is majorly composed of shear planes rather than the (010) planes preferred by Li^+ intercalation. As a result, when the desolvated Li^+ ions are prepared to enter the lattice, they mostly encounter these shear planes, posing a high energy barrier for entry into the crystal. This sluggish Li^+ intercalation at the shear planes could lead to the accumulation of more Li^+ at these surface regions, potentially causing re-solvation into the electrolyte and significantly compromising the rate capability. Recognizing that such anisotropic Li^+ transport property of NbWO host is the key bottleneck in achieving ultrafast

charging within one minute, we thus discuss in the following sections how such anisotropy and its consequences can be effectively addressed using a simple surface coating strategy. The specific changes include (added on manuscript Page 12 in the revised manuscript):

“Since our investigation focuses on the diffusion mechanism of lithium ions within the bulk of the material, *in situ* TEM was employed to observe the electrochemical lithiation process in NbWO to further investigate the limiting factors of NbWO's performance, the method of which has been widely used to study the Li⁺ storage mechanisms in various electrode materials with a high degree of reliability, such as the high consistencen between the lithiation kinetics of silicon anode studied via *in situ* solid-state TEM, *in situ* liquid TEM and *ex situ* TEM of real battery-cycled Si anode³⁵⁻³⁷. The single crystals with [010] and [101] zone axes were obtained by performing a focused ion beam (FIB) along the cross-section ([010] zone axis) and perpendicular to the cross-section ([101] zone axis, Supplementary Fig. 14). The experiments were conducted using an all-solid electrochemical cell setup, comprising a working electrode of NbWO single crystal, a naturally-grown Li₂O solid electrolyte, and a bulk lithium metal counter electrode (Fig. 3a). A bias potential of -3 V was used to initiate the lithiation of the bulk in NbWO particle.”

We hope this explanation clarifies the purpose and relevance of the *in situ* TEM data in our study.

4. rGO is a well-known material to improve the electrode performance. The authors claim that data science proposed rGO, but no quantitative analysis/discussion is provided in the main text, making this approach unconvincing. This section needs further improvement, for example, by predicting electrode performance improvement effects quantitatively using the features of physical properties (electronic conductivity, ionic conductivity, etc.).

Response to comment-4: We appreciate the reviewer' s insightful comments regarding the need for quantitative prediction of electrode performance improvements. We acknowledge that this is an important aspect to strengthen the proposed approach. In

response, we have thoroughly discussed the relevant electrochemical properties required for the rGO that enhance electrode performance. Specifically, we predicted the electronic conductivity, ionic conductivity, and oxidation potential for all materials in our dataset. From these predictions, we identified the material with the most optimal properties, which has further improved this section of the study. Additionally, we conducted precise measurements on the NbWO before and after modification. Using a four-probe method on powder pellets, we tested the I - V curves, along with *in situ* TEM testing of single micron-sized particles. These measurements confirm that the improvements extend beyond the microscopic scale to enhance the macroscopic performance of the materials. Finally, we performed *in situ* EIS tests and fitted the results for comparison between pre- and post-modification materials. The results showed that rGO significantly reduces the charge transfer resistance at the material interface.

In the revised version of the manuscript, we have added *in situ* EIS, *in situ* TEM for I - V curves of single crystals, Raman spectra to enhance the depth of our analysis. The corresponding revisions have been made in the manuscript (on manuscript Page 16) and supporting information (on supporting information Page 5) as follows:

“Fast charging performance enhancement and evaluation

As discussed above, we herein propose that the interface engineering strategy could function to relocate the desolvated lithium ions surrounding the NbWO surfaces to the openings of [010]-oriented tunnels where Li^+ intercalation is more favorably with a low energy barrier, which can overcome the last bottleneck for ultra-fast charging in NbWO materials. **Firstly, a high-throughput screening process is implemented to explore potential anode coating materials, as shown in Fig. 4a. Around 83989 structure and property data entries were obtained from the Materials Project database³⁸ (detailed calculation methods are provided in the supporting information). Each entry contains data such as the crystal structure, space group number, formation energy, and band gap of the material. In lithium-ion batteries, the electrodes are mixed conductors that require both good electronic and ionic conductivity to connect with external and internal circuits. Therefore, the electrode coating materials also need to have high electronic**

and ionic conductivity, as well as a high oxidation potential to maintain electrochemical stability. A screening process for coating materials is conducted following the acquisition of electrochemical properties using the ACGNet model and the electronic conductivity calculation formula³⁹. The initial screening eliminates unsuitable materials containing radioactive, toxic elements or precious metal elements, ensuring that the remaining candidates offer cost advantages and meet fundamental stability requirements. The dataset after initial screening is presented in Fig. 4b, where each point represents a material with its x, y, and z coordinates signifying electronic conductivity (σ_e), ionic conductivity (σ_i), and oxidation potential (V_{ox}), respectively. Further refinement, targeting a V_{ox} above 4 V, σ_e above 10^3 S cm^{-1} , and σ_i above $10^{-6} \text{ S cm}^{-1}$, identifies graphene as the most promising coating material with the optimal comprehensive properties, as marked by the red dot in Fig. 4b. Specifically, its electronic conductivity, ionic conductivity, and oxidation potential are $9.2 \times 10^3 \text{ S cm}^{-1}$, $5.61 \times 10^{-6} \text{ S cm}^{-1}$, and 5.38 V, respectively. Compared to mechanically exfoliated graphene, reduced graphene oxide (rGO) boasts a simpler preparation method and functional groups such as carboxyl and hydroxyl (Supplementary Fig. 15), facilitating its integration with organic electrolytes in lithium-ion batteries⁴⁰. rGO also retains the electrochemical characteristics of conventional graphene stemming from its crystalline structure and size effects. Based on the above analysis, rGO was selected as a potential anode coating material and conducted subsequent characterization and electrochemical experiments.

Among various interface control strategies, the target rGO considering its low desolvation energy barrier and high lithium-ion diffusion rate that can guarantee fast Li^+ transport and relocation^{12,41}. DFT theoretical calculations (Fig. 4c) show that the diffusion energy barrier of lithium ions in rGO is lower than that of migration and diffusion in NbWO tunnel structure, which indicates that coating rGO in NbWO can effectively increase the overall ionic conductivity of the material, which is conducive to the realization of significantly facilitated diffusion of lithium ions to obtain ultra-high rate performance.

Hence, the NbWO was treated with H₂ plasma to improve contact with GO and fabricate the final product of rGO/NbWO⁴². The yellow powder turns gray (Supplementary Fig. 16), indicating the uniform coating of rGO on the stick-like NbWO particle (Supplementary Fig. 4). EDS elemental mappings show a uniform distribution of C elements on the material surface (Supplementary Fig. 17). As shown in Supplementary Fig. 18, the weight loss from the adsorbed water for NbWO is around 0.7% at 1000 °C, while the weight loss of rGO/NbWO is around 1.2%. This result indicates the weight of the added rGO is about 0.5 wt.%. However, the overall peak position of XPS is shifted after coating the rGO layer, suggesting a change in the chemical state of Nb and W. Furthermore, the C 1s spectra of rGO/NbWO indicate that the material is coated with rGO, and the proportion of C=O in the C 1s peak is small and shows that GO is successfully reduced to rGO (Supplementary Fig. 19)⁴³. Meanwhile, compared to NbWO, the Raman spectrum of rGO/NbWO shows distinct D and G peaks, indicating a higher degree of graphitization on the material's surface, which suggests improved electronic conductivity (Supplementary Fig. 20).

The *I–V* responses of single NbWO and rGO/NbWO crystals, measured via *in situ* TEM (inset of Fig. 4d), are linear and symmetric in the high-bias regime (5 V). The *I–V* curves exhibit an almost linear relationship (Fig. 4d), allowing the conductance (*G*) of the crystals to be calculated using $G = dI/dV$, *G* was calculated to be $1.69 \times 10^{-3} \mu\text{S}$ for NbWO and $3.45 \times 10^{-2} \mu\text{S}$ for rGO/NbWO, indicating a significant enhancement in electronic conductivity for NbWO due to rGO presence. Additionally, the powder conductivity was tested using the four-probe method (Supplementary Fig. 21), the electronic conductivity of rGO/NbWO is calculated to be $5.50 \times 10^{-1} \text{ S m}^{-1}$, while that of the NbWO is only $2.85 \times 10^{-6} \text{ S m}^{-1}$, indicating that the electronic resistance of rGO/NbWO crystal is small.

As illustrated in Fig. 4e, without any surface modification, lithium ions on the surface of the NbWO crystal can only enter the bulk phase through the tunnel openings, while the shear planes, appearing as the major portion of particle surface area, impede the intercalation of lithium ions, resulting in poor overall lithium-ion diffusion kinetics at these surfaces. However, coating NbWO with a rGO layer reduces anisotropy at the

interface and fosters a uniform Li-rich surface layer, thereby streamlining the diffusion pathway from the shear surface to along the rGO layer, which enhances the transfer of desolvated Li^+ ions from the electrolyte to the tunnel pathways^{47,48}. As shown in Supplementary Fig. 22, *in situ* electrochemical impedance spectroscopy (EIS) measurements demonstrate that rGO/NbWO exhibits lower intrinsic impedance (3.83–4.35 Ω) and significantly reduced lithium-ion desolvation impedance at the interface compared to NbWO. Consequently, the enhanced electronic conduction and Li^+ transfer in the interfacial region facilitate a more uniform Li-(de)intercalation layer, fostering synchronized electrochemical reactions.

Fig. 4. Fast charging performance enhancement. **a** The overall screening procedure from the Materials Project database. **b** Initial screening results of candidate coating materials under 3 selected dimensions: electronic conductivity, ionic conductivity, and oxidation potential. **c** Minimum energy paths and activation barriers (NEB) for lithium-ion motion in rGO/NbWO. **d** I-V curves of NbWO and rGO/NbWO particles measured

in situ under the microscope. **e** Schematic illustrations of lithiation processes of NbWO₃ and rGO/NbWO₃.

Supporting Information

Machine learning-assisted interface engineering strategy: Coating materials generally need to prioritize meeting relevant requirements such as high electronic and ionic conductivity, as well as a high oxidation potential to maintain electrochemical stability. Therefore, we first used the ACGNet model to predict the ionic conductivity σ_i and oxidation potential V_{ox} (versus Li⁺/Li) of each material. ACGNet is a deep learning model based on graph neural networks that can predict the corresponding properties of input material crystal structures. The training data of ACGNet comes from the dataset provided by Austin D. Sendek et al². The dataset includes the crystal structures of 40 crystalline materials and their bulk ionic conductivities at room temperature, as well as the crystal structures and oxidation potentials of 216 other materials. Subsequently, the electronic conductivity of each material was approximated using a conductivity calculation formula based on carrier mobility and volume concentration³. The electronic conductivity for an intrinsic crystalline semiconductor can be approximated as:

$$\sigma_e = (\mu_e + \mu_h)q\sqrt{N_c N_v}e^{-E_{gap}/2kT} \quad (1)$$

Where μ_e and μ_h denote the electron and hole mobilities, q is the charge of an electron, N_c and N_v represent the state densities in the conduction and valence bands, respectively. The standard values for crystalline silicon ($N_c = 2.89 \times 10^{19} \text{ cm}^{-3}$, $N_v = 3.13 \times 10^{19} \text{ cm}^{-3}$, $\mu_e = 1430 \text{ cm}^2 \text{ V}^{-1} \text{ s}^{-1}$, $\mu_h = 480 \text{ cm}^2 \text{ V}^{-1} \text{ s}^{-1}$, $T = 300 \text{ K}$) is employed to derive the electronic conductivity of materials at room temperature. Although the parameters in Equation (1) vary across materials, the general trend of a larger bandgap leading to lower electronic conductivity remains consistent. Consequently, this formula, in conjunction with the bandgap of each material, is utilized to crudely estimate the electronic conductivity of each material.

Supplementary Fig. 15. The FTIR spectrum of rGO/Nb₁₆W₅O₅₅.

Supplementary Fig. 20. The Raman spectra of Nb₁₆W₅O₅₅ and rGO/Nb₁₆W₅O₅₅.

Supplementary Fig. 21. The *I-V* curves of Nb₁₆W₅O₅₅ and rGO/Nb₁₆W₅O₅₅ tested using the four-probe method.

Supplementary Fig. 22. *In situ* EIS for Nb₁₆W₅O₅₅ and rGO/Nb₁₆W₅O₅₅ over a voltage range of 1 V to 3 V.”

5. Although the authors discuss an interesting solid-state electrochemistry of Nb₁₆W₅O₅₅ in the first half of the manuscript, which I believe is worthy of publication in Nat. Commun., the other half focuses on the development of rGO/Nb₁₅W₅O₅₅ composite and full cell, which were conducted routinely. I can find many similar papers in many journals, which are not relevant to either basic science or practical battery technology. Therefore, I strongly recommend to thoroughly rewrite the manuscript by focusing on the rate-dependent electrochemistry of Nb₁₆W₅O₅₅.

Response to Comment-5: Thank you for this comment. As you said, rGO is a common composite material and in most of the previous reports this strategy was also proposed and confirmed the effectiveness of rGO in interfacial engineering. In this work, we also hope to address the above challenges from simple interfacial engineering, although it is not the central focus of our study. We have rationalized the application of rGO according to your fourth comment. In addition, we have included an additional set of DFT calculations for 22 different structures, covering both homogeneous and heterogeneous intercalation models. These calculations further validate the voltage states of the material under both low and high-rate conditions, showing strong agreement with the experimental results. At the same time, we have rearranged the logic of the section, and all revisions are marked in red in the manuscript. We hope that the revised manuscript will be more logically understandable to readers.

Reviewer #3:

The paper has a very general title which piqued my interest. However, the content turned out to be very, very specific and related to detailed aspects of a single material. My first suggestion would be to change the title, otherwise readers may also be confused. Being a more general materials scientist/electrochemist, I had some difficulty in assessing the paper due to the wealth of detailed information. I suggest that some dedicated experts should also review this paper. In terms of analytical tools, the paper seems to be at a very high level. However, I have some general comments and questions. They largely address the discussion of the results (for the moment I do not see the need for more experiments). If the points are properly addressed, I think the paper can be a very valuable contribution to the battery research community.

Response to Reviewer 3: Thank you very much for your thoughtful comment and for highlighting the importance of aligning the title with the content of the paper. We understand that the previous title may have been too general, which could lead to confusion. To better reflect the specific focus of our study on Nb₁₆W₅O₅₅-based materials, we have revised the title to “**Make NbWO-based fast-charging batteries charging faster**”. Thank you again for your valuable suggestions, which we believe will contribute to the quality and impact of our work in the battery research community.

1. Authors identify rGO as the most promising coating substance. A better description of rGO should be provided. rGO is generally a very ill-defined material so it remains unclear why it comes out as first choice. What is the actual reason? How does the applied rGO compares to the structure of the rGO from the materials database?

Response to Comment-1: Thank you for the insightful comment regarding the description of reduced graphene oxide (rGO). In the original manuscript, we may have failed to fully explain the advantages of rGO as a coating material and its specific application in our study. Graphene is a two-dimensional material with a high specific surface area, exceptional electrical conductivity, and excellent mechanical strength. These properties make graphene highly promising in fields like electrochemical energy

storage, catalyst support, and composite material enhancement. Its high conductivity can effectively reduce internal electrode resistance and enhance charge transport efficiency, which is critical for improving electrochemical performance.

Why rGO comes out as first choice? We employed a deep-learning model to predict the electronic conductivity, ionic conductivity, and oxidation potential of all materials in the dataset. We then screened the materials based on the following criteria: oxidation potential greater than 4 V, ionic conductivity greater than 10^{-3} S/cm, and electronic conductivity greater than 10^6 S/cm. Graphene was selected as the coating material due to its superior electronic conductivity, relatively good ionic conductivity, and high oxidation potential.

In the revised version of the manuscript, we further carried out *in situ* EIS, *in situ* TEM for *I-V* curves of single crystals and Raman spectra to explore the key electrochemical properties of the anode materials before and after rGO addition, which we believe significantly deepened our understanding of the role of rGO in this system.

Comparison of rGO with Graphene from the Database: Reduced graphene oxide (rGO) shares the same crystal structure and size effects as the graphene in the database, resulting in similar electrochemical properties. Additionally, rGO offers a more mature preparation method and lower cost. Its surface, rich in organic functional groups, can better interact with the organic electrolyte in lithium-ion batteries, enhancing interfacial transport properties. Therefore, we chose rGO for further characterization and experiments in this study.

Compared to mechanical exfoliation, the rGO has the advantage of producing graphene in large quantities at a relatively low cost. However, the oxidation-reduction process may introduce trace oxygen-containing groups, slightly affecting the physical and chemical properties of graphene. Nonetheless, rGO retains similar physicochemical properties to standard graphene, with a consistent crystal structure, and is equally effective in enhancing the conductivity and rate performance of electrodes.

To improve this, we have added the following content to provide a more comprehensive description of rGO on manuscript Page 16 and supporting information (on supporting information Page 5):

“Fast charging performance enhancement and evaluation

As discussed above, we herein propose that the interface engineering strategy could function to relocate the desolvated lithium ions surrounding the NbWO surfaces to the openings of [010]-oriented tunnels where Li^+ intercalation is more favorably with a low energy barrier, which can overcome the last bottleneck for ultra-fast charging in NbWO materials. Firstly, a high-throughput screening process is implemented to explore potential anode coating materials, as shown in Fig. 4a. Around 83989 structure and property data entries were obtained from the Materials Project database³⁸ (detailed calculation methods are provided in the supporting information). Each entry contains data such as the crystal structure, space group number, formation energy, and band gap of the material. In lithium-ion batteries, the electrodes are mixed conductors that require both good electronic and ionic conductivity to connect with external and internal circuits. Therefore, the electrode coating materials also need to have high electronic and ionic conductivity, as well as a high oxidation potential to maintain electrochemical stability. A screening process for coating materials is conducted following the acquisition of electrochemical properties using the ACGNet model and the electronic conductivity calculation formula³⁹. The initial screening eliminates unsuitable materials containing radioactive, toxic elements or precious metal elements, ensuring that the remaining candidates offer cost advantages and meet fundamental stability requirements. The dataset after initial screening is presented in Fig. 4b, where each point represents a material with its x, y, and z coordinates signifying electronic conductivity (σ_e), ionic conductivity (σ_i), and oxidation potential (V_{ox}), respectively. Further refinement, targeting a V_{ox} above 4 V, σ_e above 10^3 S cm^{-1} , and σ_i above $10^{-6} \text{ S cm}^{-1}$, identifies graphene as the most promising coating material with the optimal comprehensive properties, as marked by the red dot in Fig. 4b. Specifically, its electronic conductivity, ionic conductivity, and oxidation potential are $9.2 \times 10^3 \text{ S cm}^{-1}$, $5.61 \times 10^{-6} \text{ S cm}^{-1}$, and 5.38 V, respectively. Compared to mechanically exfoliated graphene, reduced graphene oxide (rGO) boasts a simpler preparation method and functional groups such as carboxyl and hydroxyl (Supplementary Fig. 15), facilitating its integration with organic electrolytes in lithium-ion batteries⁴⁰. rGO also retains the

electrochemical characteristics of conventional graphene stemming from its crystalline structure and size effects. Based on the above analysis, rGO was selected as a potential anode coating material and conducted subsequent characterization and electrochemical experiments.

Among various interface control strategies, the target rGO considering its low desolvation energy barrier and high lithium-ion diffusion rate that can guarantee fast Li^+ transport and relocation^{12,41}. DFT theoretical calculations (Fig. 4c) show that the diffusion energy barrier of lithium ions in rGO is lower than that of migration and diffusion in NbWO tunnel structure, which indicates that coating rGO in NbWO can effectively increase the overall ionic conductivity of the material, which is conducive to the realization of significantly facilitated diffusion of lithium ions to obtain ultra-high rate performance.

Hence, the NbWO was treated with H_2 plasma to improve contact with GO and fabricate the final product of rGO/NbWO⁴². The yellow powder turns gray (Supplementary Fig. 16), indicating the uniform coating of rGO on the stick-like NbWO particle (Supplementary Fig. 4). EDS elemental mappings show a uniform distribution of C elements on the material surface (Supplementary Fig. 17). As shown in Supplementary Fig. 18, the weight loss from the adsorbed water for NbWO is around 0.7% at 1000 °C, while the weight loss of rGO/NbWO is around 1.2%. This result indicates the weight of the added rGO is about 0.5 wt.%. However, the overall peak position of XPS is shifted after coating the rGO layer, suggesting a change in the chemical state of Nb and W. Furthermore, the C 1s spectra of rGO/NbWO indicate that the material is coated with rGO, and the proportion of C=O in the C 1s peak is small and shows that GO is successfully reduced to rGO (Supplementary Fig. 19)⁴³. Meanwhile, compared to NbWO, the Raman spectrum of rGO/NbWO shows distinct D and G peaks, indicating a higher degree of graphitization on the material's surface, which suggests improved electronic conductivity (Supplementary Fig. 20).

The I - V responses of single NbWO and rGO/NbWO crystals, measured via *in situ* TEM (inset of Fig. 4d), are linear and symmetric in the high-bias regime (5 V). The I - V curves exhibit an almost linear relationship (Fig. 4d), allowing the conductance (G)

of the crystals to be calculated using $G = dI/dV$, G was calculated to be $1.69 \times 10^{-3} \mu\text{S}$ for NbWO and $3.45 \times 10^{-2} \mu\text{S}$ for rGO/NbWO, indicating a significant enhancement in electronic conductivity for NbWO due to rGO. Additionally, the powder conductivity was tested using the four-probe method (Supplementary Fig. 21), the electronic conductivity of rGO/NbWO is calculated to be $5.50 \times 10^{-1} \text{ S m}^{-1}$, while that of the NbWO is only $2.85 \times 10^{-6} \text{ S m}^{-1}$, indicating that the electronic resistance of rGO/NbWO crystal is small.

As illustrated in Fig. 4e, without any surface modification, lithium ions on the surface of the NbWO crystal can only enter the bulk phase through the tunnel openings, while the shear planes, appearing as the major portion of particle surface area, impede the intercalation of lithium ions, resulting in poor overall lithium-ion diffusion kinetics at these surfaces. However, coating NbWO with a rGO layer reduces anisotropy at the interface and fosters a uniform Li-rich surface layer, thereby streamlining the diffusion pathway from the shear surface to along the rGO layer, which enhances the transfer of desolvated Li^+ ions from the electrolyte to the tunnel pathways^{47,48}. As shown in Supplementary Fig. 22, *in situ* electrochemical impedance spectroscopy (EIS) measurements demonstrate that rGO/NbWO exhibits lower intrinsic impedance (3.83–4.35 Ω) and significantly reduced lithium-ion desolvation impedance at the interface compared to NbWO. Consequently, the enhanced electronic conduction and Li^+ transfer in the interfacial region facilitate a more uniform Li-(de)intercalation layer, fostering synchronized electrochemical reactions.

Fig. 4. Fast charging performance enhancement. **a** The overall screening procedure from the Materials Project database. **b** Initial screening results of candidate coating materials under 3 selected dimensions: electronic conductivity, ionic conductivity, and oxidation potential. **c** Minimum energy paths and activation barriers (NEB) for lithium-ion motion in rGO/NbWO. **d** I-V curves of NbWO and rGO/NbWO particles measured *in situ* under the microscope. **e** Schematic illustrations of lithiation processes of NbWO and rGO/NbWO.

Machine learning-assisted interface engineering strategy: Coating materials generally need to prioritize meeting relevant requirements such as high electronic and ionic conductivity, as well as a high oxidation potential to maintain electrochemical stability. Therefore, we first used the ACGNet model to predict the ionic conductivity σ_i and oxidation potential V_{ox} (versus Li^+/Li) of each material¹. ACGNet is a deep

learning model based on graph neural networks that can predict the corresponding properties of input material crystal structures. The training data of ACGNet comes from the dataset provided by Austin D. Sendek et al². The dataset includes the crystal structures of 40 crystalline materials and their bulk ionic conductivities at room temperature, as well as the crystal structures and oxidation potentials of 216 other materials. Subsequently, the electronic conductivity of each material was approximated using a conductivity calculation formula based on carrier mobility and volume concentration³. The electronic conductivity for an intrinsic crystalline semiconductor can be approximated as:

$$\sigma_e = (\mu_e + \mu_h)q\sqrt{N_c N_v}e^{-E_{gap}/2kT} \quad (1)$$

Where μ_e and μ_h denote the electron and hole mobilities, q is the charge of an electron, N_c and N_v represent the state densities in the conduction and valence bands, respectively. The standard values for crystalline silicon ($N_c = 2.89 \times 10^{19} \text{ cm}^{-3}$, $N_v = 3.13 \times 10^{19} \text{ cm}^{-3}$, $\mu_e = 1430 \text{ cm}^2 \text{ V}^{-1} \text{ s}^{-1}$, $\mu_h = 480 \text{ cm}^2 \text{ V}^{-1} \text{ s}^{-1}$, $T=300 \text{ K}$) is employed to derive the electronic conductivity of materials at room temperature. Although the parameters in Equation (1) vary across materials, the general trend of a larger bandgap leading to lower electronic conductivity remains consistent. Consequently, this formula, in conjunction with the bandgap of each material, is utilized to crudely estimate the electronic conductivity of each material.

Supplementary Fig. 15. The FTIR spectrum of rGO/Nb₁₆W₅O₅₅.

Supplementary Fig. 20. The Raman spectra of $\text{Nb}_{16}\text{W}_5\text{O}_{55}$ and $\text{rGO}/\text{Nb}_{16}\text{W}_5\text{O}_{55}$.

Supplementary Fig. 21. The I - V curves of $\text{Nb}_{16}\text{W}_5\text{O}_{55}$ and $\text{rGO}/\text{Nb}_{16}\text{W}_5\text{O}_{55}$ tested using the four-probe method.

Supplementary Fig. 22. *In situ* EIS for $\text{Nb}_{16}\text{W}_5\text{O}_{55}$ and $\text{rGO}/\text{Nb}_{16}\text{W}_5\text{O}_{55}$ over a voltage range of 1 V to 3 V.”

2. It is not clearly described enough, why rGO is effective in improving the anisotropy of the particles. The explanation on page 16/17 sounds very general and seems a hypothesis only. But is there any evidence?

Response to comment-2: Thank you for your valuable comment regarding the role of rGO. We appreciate the opportunity to clarify this aspect. Our primary objective was to enhance the lithium-ion migration pathways at the material interface. The presence of rGO provides a diffusion pathways within the rGO layers on the surface of the NbWO. This modification reduced the desolvation impedance of lithium ions at the material interface. This conclusion is supported by DFT calculations, which demonstrated a reduction in the lithium-ion energy barrier, and also supported by *in situ* EIS measurements, which confirmed a decrease in the desolvation energy barrier at the material's surface. Additionally, we conducted *in situ* TEM observations for the *I-V* curves of single crystals, and employed the four-probe method to obtain the *I-V* curves for pressed pellets, further emphasizing the improvements in electronic conductivity due to interface modification.

In the revised version of the manuscript, we have added *in situ* EIS, *in situ* TEM for *I-V* curves of single crystals, Raman spectra in our manuscript to enhance the depth of our analysis. The specific revisions are as follows (On manuscript Page 17):

“DFT theoretical calculations (Fig. 4c) show that the diffusion energy barrier of lithium ions in rGO is lower than that of migration and diffusion in NbWO tunnel structure, which indicates that coating rGO in NbWO can effectively increase the overall ionic conductivity of the material, which is conducive to the realization of significantly facilitated diffusion of lithium ions to obtain ultra-high rate performance.”

“The *I-V* responses of NbWO and rGO/NbWO crystals, measured *in situ* under the microscope (inset of Fig. 4d), are linear and symmetric in the high-bias regime (5 V). The *I-V* curves exhibit an almost linear relationship (Fig. 4d), allowing the conductance (G) of the crystals to be calculated using $G = dI/dV$, G was calculated to be $1.69 \times 10^{-3} \mu\text{S}$ for NbWO and $3.45 \times 10^{-2} \mu\text{S}$ for rGO/NbWO, indicating a significant enhancement in electronic conductivity for NbWO due to rGO. Additionally, the powder conductivity was tested using the four-probe method (Supplementary Fig. 21),

the electronic conductivity of rGO/NbWO is calculated to be $5.50 \times 10^{-1} \text{ S m}^{-1}$, while that of the NbWO is only $2.85 \times 10^{-6} \text{ S m}^{-1}$, indicating that the electronic resistance of rGO/NbWO crystal is small.

As illustrated in Fig. 4e, without any surface modification, lithium ions on the surface of the NbWO crystal can only enter the bulk phase through the tunnel openings, while the shear planes, appearing as the major portion of particle surface area, impede the intercalation of lithium ions, resulting in poor overall lithium-ion diffusion kinetics at these surfaces. However, coating NbWO with a rGO layer reduces anisotropy at the interface and fosters a uniform Li-rich surface layer, thereby streamlining the diffusion pathway from the shear surface to along the rGO layer, which enhances the transfer of desolvated Li^+ ions from the electrolyte to the tunnel pathways^{47,48}. As shown in Supplementary Fig. 22, *in situ* electrochemical impedance spectroscopy (EIS) measurements demonstrate that rGO/NbWO exhibits lower intrinsic impedance (3.83–4.35 Ω) and significantly reduced lithium-ion desolvation impedance at the interface compared to NbWO. Consequently, the enhanced electronic conduction and Li^+ transfer in the interfacial region facilitate a more uniform Li-(de)intercalation layer, fostering synchronized electrochemical reactions.

Supplementary Fig. 20. The Raman spectra of $\text{Nb}_{16}\text{W}_5\text{O}_{55}$ and $\text{rGO}/\text{Nb}_{16}\text{W}_5\text{O}_{55}$.

Supplementary Fig. 21. The I - V curves of $\text{Nb}_{16}\text{W}_5\text{O}_{55}$ and $\text{rGO}/\text{Nb}_{16}\text{W}_5\text{O}_{55}$ tested using the four-probe method.

Supplementary Fig. 22. *In situ* EIS for $\text{Nb}_{16}\text{W}_5\text{O}_{55}$ and $\text{rGO}/\text{Nb}_{16}\text{W}_5\text{O}_{55}$ over a voltage range of 1 V to 3 V.”

3. Authors state values for the specific capacity with an extra digit, e.g. 257.2 mAh/g. But is this reproducible? Authors may round number to e.g. 257 mAh/g.

Response to comment 3: Thank you for your observation regarding the specific capacity values reported in the manuscript. To clarify, each set of experiments was repeated eight times to ensure accuracy and reliability, and the results consistently demonstrated high reproducibility. However, we understand the importance of presenting data in a clear and standardized manner. Therefore, we revise the manuscript to round the specific capacity values, such as changing "257.2 mAh g⁻¹" to "257 mAh g⁻¹". This adjustment will make the data more accessible while still accurately reflecting our findings. Specific revisions are as follows:

[1] To overcome such anisotropy, we propose a machine learning-assisted interface engineering strategy to swiftly collect desolvated Li^+ and relocate them to (010) surfaces for their fast intercalation. As a result, a capacity of $\approx 116 \text{ mAh g}^{-1}$ (68.5% of the theoretical capacity) at 80 C is achieved when coupled with a Li anode. (On manuscript Page 2)

[2] Hence, it resulted in a significant improvement in high-rate performance ($\approx 116 \text{ mAh g}^{-1}$ at 80 C), far surpassing traditional $\text{Nb}_{16}\text{W}_5\text{O}_{55}$ (which exhibits a capacity below 50 mAh g^{-1} at 60 C), and meets the fast-charging criteria outlined by USABC (United States Advanced Battery Consortium defines fast charging as reaching 80% State of Charge (SOC) within 15 minutes (4C rate), and rGO/NbWO achieves this even at nearly 40C). (On manuscript Page 4)

[3] Applying an rGO coating creates a lithium-rich layer, facilitating the rapid transfer of Li^+ ions to tunnel openings. This enhancement improves high-rate performance, achieving 257 mAh g^{-1} at 0.2 C, 168 mAh g^{-1} at 10 C, and 116 mAh g^{-1} at 80 C. (On manuscript Page 20)

[4] In addition, the specific discharge capacity of $\text{LiFePO}_4\|\text{rGO/NbWO}$ at 5 C was 85 mAh g^{-1} and remained 96.1% at the first 100 cycles. When the pouch cell was fast-charged and discharged for 500 cycles at 5 C, the capacity could still be maintained at around 77.0% (Fig. 5g). (On manuscript Page 21)

[5] As a result, the rGO/NbWO exhibits a capacity of 257 mAh g^{-1} at 0.2 C and 116 mAh g^{-1} at 80 C, which maintains a capacity of 92.7% for 500 cycles at a rate of 10 C. (On manuscript Page 22)

4. The authors correctly mention in the introduction that the nanosizing of Si anodes (or Si/carbon anodes) leads to a penalty in volumetric energy density. Authors should comment on the volumetric energy density (and gravimetric energy density) of their material and cells.

Response to comment-4: Thank you for your insightful comment. Investigating low-cost, high-tap-density, and high-volumetric-energy-density micron-sized materials with low specific surface area (to reduce side reactions and improve initial Coulombic

efficiency) is indeed critical, as highlighted in ACS Nano 2021, 15, 15567–15593. To emphasize the fast-charging advantage of the micron-sized rGO/Nb₁₆W₅O₅₅ in this study, we compared its performance with advanced NbWO series materials. Our results show that rGO/Nb₁₆W₅O₅₅, with a tap density of 1.61 g cm⁻³, can achieve an impressive energy density of 186 Wh kg⁻¹ even at a high power density of 14,860 W kg⁻¹, and 406 Wh kg⁻¹ at 81 W kg⁻¹. Moreover, this micron-sized material offers the potential for fabricating thicker electrodes, which could further enhance both gravimetric and volumetric energy densities. We also evaluated the full cell performance using the BatPac model, simulating a 100 Ah LFP||rGO/Nb₁₆W₅O₅₅ cell. The results indicate a volumetric energy density of 251 Wh L⁻¹ and a gravimetric energy density of 123 Wh kg⁻¹. Based on this, we have made the following revisions to the manuscript (On manuscript Page 23):

“Fig. 5h shows that rGO/Nb₁₆W₅O₅₅ electrode, with a tap density of 1.61 g cm⁻³, can achieve an impressive energy density of 186 Wh kg⁻¹ at a high power density of 14,860 W kg⁻¹, and 406 Wh kg⁻¹ at 81 W kg⁻¹, surpassing most of NbWO based materials. Furthermore, using the BatPac model developed by Argonne National Laboratory^{50,51}, we simulated a 100 Ah LFP||rGO/Nb₁₆W₅O₅₅ cell, which demonstrated a volumetric energy density of 251 Wh L⁻¹ and a gravimetric energy density of 123 Wh kg⁻¹, suggesting promising potential for practical applications.

Fig. 5. Fast charging performance evaluation. a Rate performance for rGO/NbWO and NbWO. **b** Comparison of rate performance in different works of literature. **c** Nyquist plot and fitting data for rGO/NbWO and NbWO. **d** dQ/dV plots of rGO/NbWO in 1 M LiTFSI. **e** Cycling performance of rGO/NbWO coin cells at 1 C and 10 C. **f** Schematic diagram of pouch cell. **g** Cycling performance of LiFePO₄||rGO/NbWO pouch cell at 5 C. **h** Energy density and power density of NbWO based materials.”

5. Generally, fast charging can be obtained by decreasing the areal loading of the electrode. Authors should comment on that and discuss their results. What is the areal capacity compared to conventional high-rate batteries (mAh/cm²)?

Response to comment-5: Thank you for your comment. Typically, fast-charge performance is evaluated using coin cells with low areal loadings, which enhance lithium-ion diffusion and electron conduction. While this approach improves rate

performance, it does not reflect real-world applications, where higher areal loadings are required to achieve greater energy densities. Therefore, we evaluated the electrochemical performance under different loading conditions.

As rGO/NbWO is a fast-charging anode material, most performance tests were conducted at low loading levels (1~3 mg cm⁻²), yielding an areal capacity of around 0.44 mAh cm⁻². However, as the loading increases, the rate performance deteriorates significantly. For example, at a loading of 15 mg cm⁻², the impedance rises to 198 Ω, and the capacity reaches 208 mAh g⁻¹ at 0.2 C but drops to 24 mAh g⁻¹ at 40 C (Supplementary Fig. 28), with the areal capacity reaching 2.88 mAh cm⁻². In comparison, the real capacity of conventional fast-charging LFP||LTO systems is only around 1.1 mAh cm⁻² (*Nat. Commun.* 15, 6299, 2024), indicating the promising potential of rGO/NbWO for practical applications.

We have made the following revisions to the manuscript accordingly on manuscript Page 23.

“However, the current performance in practical applications is not ideal, primarily due to several limiting factors. As rGO/NbWO is a fast-charging anode material, most performance evaluations have been conducted with low loading levels (1~3 mg cm⁻²). When the loading increases, its rate performance deteriorates significantly. For instance, the rGO/NbWO with an areal capacity of around 0.44 mAh cm⁻² achieves 257 mAh g⁻¹ at 0.2 C, when the loading is increased to 15 mg cm⁻² with the areal capacity reaching 2.88 mAh cm⁻², the impedance rises to 198 Ω, and the capacity is 208 mAh g⁻¹ at 0.2 C and drops to 24 mAh g⁻¹ at 40 C (Supplementary Fig. 28). Additionally, the commercial LFP shows a significant decline in rate performance under high current conditions, which results in the relatively low capacity of the LiFePO₄||rGO/NbWO pouch cell (Supplementary Fig. 29). Fig. 5h result shows that rGO/Nb₁₆W₅O₅₅ with a tap density of 1.61 g cm⁻³, can achieve an impressive energy density of 186 Wh kg⁻¹ at a high power density of 14,860 W kg⁻¹, and 406 Wh kg⁻¹ at 81 W kg⁻¹, super than most of NbWO based materials. Furthermore, using the BatPac model developed by Argonne National Laboratory^{50,51}, we simulated a 100 Ah LFP||rGO/Nb₁₆W₅O₅₅ cell, which demonstrated

a volumetric energy density of 251 Wh L⁻¹ and a gravimetric energy density of 123 Wh kg⁻¹, suggesting promising potential for practical applications.

Supplementary Fig. 28. The a) EIS and b) the rate performance of rGO/Nb₁₆W₅O₅₅ with different mass loading.

Supplementary Fig. 29. The a) rate performance and b) the cycling performance of the LFP cathode.”

6. Authors motivate their study clearly by application (military and space applications). So I miss a balanced discussion on energy density and fast charging. For example, also a plot, where existing high-rate batteries are compared to the cell the authors make. Maybe a Ragone plot? From the provided information, I could not yet see how relevant the findings are in practice. The electrode of the authors also contains 15 wt% carbon additive and 5 wt% binder. This is OK for research purposes but if authors claim relevance for the application it should be compared to the composition of state-of-art anodes.

Response to comment-6: Thank you for your valuable comment. Since fast-charging

applications are particularly relevant for electric vehicles and fast-charging smartphones mostly, we have included a Ragone plot (Fig 5h) to further discuss the energy and power density of our material. Our results show that rGO/Nb₁₆W₅O₅₅, with a tap density of 1.61 g cm⁻³, can achieve an impressive energy density of 186 Wh kg⁻¹ at a high power density of 14,860 W kg⁻¹, and 406 Wh kg⁻¹ at 81 W kg⁻¹. Furthermore, using the BatPac model developed by Argonne National Laboratory, we simulated a 100 Ah LFP||rGO/Nb₁₆W₅O₅₅ cell, which demonstrated a volumetric energy density of 251 Wh L⁻¹ and a gravimetric energy density of 123 Wh kg⁻¹, suggesting promising potential for practical applications.

We primarily fabricated coin cells to observe fast-charging performance. We also compared the conductivity of cells with 15% and 10% carbon content (as shown in Response Fig. 1) and found negligible differences. However, in our pouch cell fabrication, we used a lower-carbon-content formulation, specifically with an anode composition of 94:1.5:1.5:3 (metal oxide, Super P, CNT, and PVDF, respectively), and the areal density of the anode was controlled at 6 ± 0.2 mg cm⁻² on Cu foil based on charge balance calculations. This is in line with the composition of state-of-the-art anodes used in practical applications.

Response Fig. 1 The I-V curves of rGO/Nb₁₆W₅O₅₅ with 15% carbon/5% PVDF and 10% carbon/10% PVDF for coin cells.

We have made the necessary revisions to the manuscript to reflect this comparison

on manuscript Page 23:

“In this case, the electrodes were prepared with commercial LiFePO_4 powders: Super P: CNT: PVDF = 94:1.5:1.5:3 w% as the cathode ($7.3 \pm 0.2 \text{ mg cm}^{-2}$ for each side and cathode loads on two sides) and rGO/NbWO as the anode in the same way (containing a metal oxide, Super P, CNT, and PVDF with a mass ratio of 94:1.5:1.5:3, $6 \pm 0.2 \text{ mg cm}^{-2}$ for each side and anode loads on two sides). 0.1 Ah pouch cells contain three layers of cathode and four layers of anode electrodes and constructed $\text{LiFePO}_4||\text{rGO/NbWO}$ pouch cells. In addition, the specific discharge capacity of $\text{LiFePO}_4||\text{rGO/NbWO}$ at 5 C was 85 mAh g^{-1} and remained 96.1% at the first 100 cycles. When the pouch cell was fast-charged and discharged for 500 cycles at 5 C, the capacity could still be maintained at around 77.0% (Fig. 5g). However, the current performance in practical applications is not ideal, primarily due to several limiting factors. As rGO/NbWO is a fast-charging anode material, most performance evaluations have been conducted with low loading levels ($1\sim 3 \text{ mg/cm}^2$). When the loading increases, its rate performance deteriorates significantly. For instance, the rGO/NbWO with an areal capacity of around 0.44 mAh cm^{-2} achieves 257 mAh g^{-1} at 0.2 C, when the loading is increased to 15 mg cm^{-2} with the areal capacity reaching 2.88 mAh cm^{-2} , the impedance rises to 198Ω , and the capacity is 208 mAh g^{-1} at 0.2 C and drops to 24 mAh g^{-1} at 40 C (Supplementary Fig. 28). Additionally, the commercial LFP shows a significant decline in rate performance under high current conditions, which results in the relatively low capacity of the $\text{LiFePO}_4||\text{rGO/NbWO}$ pouch cell (Supplementary Fig. 29). Fig. 5h result shows that rGO/ $\text{Nb}_{16}\text{W}_5\text{O}_{55}$ with a tap density of 1.61 g cm^{-3} , can achieve an impressive energy density of 186 Wh kg^{-1} at a high power density of $14,860 \text{ W kg}^{-1}$, and 406 Wh kg^{-1} at 81 W kg^{-1} , super than most of NbWO based materials. Furthermore, using the BatPac model developed by Argonne National Laboratory^{50,51}, we simulated a 100 Ah LFP||rGO/ $\text{Nb}_{16}\text{W}_5\text{O}_{55}$ cell, which demonstrated a volumetric energy density of 251 Wh L^{-1} and a gravimetric energy density of 123 Wh kg^{-1} , suggesting promising potential for practical applications.”

Supplementary Fig. 28. The a) EIS and b) the rate performance of rGO/Nb₁₆W₅O₅₅ with different mass loading.

Supplementary Fig. 29. The a) rate performance and b) the cycling performance of the LFP cathode.

7. I also wonder under which conditions the anode is rate limiting. Is the LFP cathode always better than the anode the authors made? Or is the prepared anode faster than the LFP cathode? Authors may also mention that rate performance is also highly temperature dependent. Maybe it is too difficult to clarify all these questions but authors should at least mention and discuss these points.

Response to comment-7: Thank you for raising these important points regarding the rate-limiting aspects of the anode and cathode in our study. We acknowledge that clarifying these issues is essential for a comprehensive understanding of the battery's performance.

In the Li||rGO/NbWO half-cell, the performance limitations under ultra-high rates are not solely attributed to the NbWO material. As a comparison, we cycled a Li||Li

symmetric cell at current densities corresponding to the 0.2C-80C rates used in Fig. 5a. The overpotential observed in the symmetric cell closely matched the overpotential seen in the electrochemical cycling curves of Supplementary Fig. 22. This indicates that near room temperature, the limiting factor in fast-charging is largely due to lithium metal plating/stripping or lithium-ion desolvation and transport in the electrolyte, rather than the anode materials. This was partially explained in our earlier report (Materials Today Energy 43, 101571, 2024)

Fast-charging batteries imply that both the anodes and cathodes possess fast-charging capability and ensure the kinetic/thermodynamic stability of the electrode/electrolyte interface during charging and discharging. Currently, phosphate-based materials such as LiFePO₄ (LFP) and Li₃V₂(PO₄)₃ cathode materials and spinel-type materials are extremely promising candidates for fast-charging Li-ion batteries. However, regarding the full-cell configuration of LFP||rGO/NbWO, the commercial LFP cathode we used showed some limitations in ultra-high-rate electrochemical performance. Despite this, our primary focus in this work was on the anode material, which is why the performance of the cathode was not elaborated in detail. In future work, developing ultra-high-rate cathode materials to pair with our anode could lead to further advancements in practical, high-rate applications.

Concerning temperature's influence on material performance, the main factor is the lithium-ion diffusion rate in the electrolyte. At higher temperatures, lithium-ion diffusion is faster, and desolvation impedance is lower, leading to better electrochemical performance. However, since most practical applications occur at room temperature, our discussion primarily focuses on these conditions. Therefore, we have revised the manuscript to reflect these considerations on manuscript Page 21 and Page 23 as follows:

“In the Li||rGO/NbWO cell, the performance limitations under ultra-high rates are not solely attributed to the NbWO material. The ionic conductivity of the electrolytes at different temperatures is presented in Supplementary Fig. 24. It was observed that the logarithm of the lithium-ion conductivity exhibits a strong linear correlation with $1000/T$, and the activation energy (E_a) of 1 M LiTFSI was calculated as 0.0345 eV by

the Arrhenius equation⁴⁹. As a comparison, we cycled a Li||Li symmetric cell at current densities corresponding to the 0.2 C-80 C rates used in Fig. 5a (Supplementary Fig. 25). The overpotential observed in the symmetric cell closely matched the overpotential seen in the electrochemical cycling curves of Supplementary Fig. 23. This indicates that near room temperature, the limiting factor in fast-charging is largely due to lithium metal plating/stripping or lithium-ion desolvation and transport in the electrolyte, rather than the anode materials.

Supplementary Fig. 23. The charging/discharging curves of rGO/Nb₁₆W₅O₅₅ in 1 M LiTFSI.

Supplementary Fig. 24. The EIS curves for Li||Li symmetrical cell and the temperature dependence curves of the ionic conductivities for different electrolytes based on stainless steel||stainless steel symmetrical cell for 1 M LiTFSI.”

Supplementary Fig. 25. Rate performance of Li||Li symmetric cell.

“In this case, the electrodes were prepared with commercial LiFePO_4 powders: Super P: CNT: PVDF = 94:1.5:1.5:3 w% as the cathode ($7.3 \pm 0.2 \text{ mg cm}^{-2}$ for each side and cathode loads on two sides) and rGO/NbWO as the anode in the same way (containing a metal oxide, Super P, CNT, and PVDF with a mass ratio of 94:1.5:1.5:3, $6 \pm 0.2 \text{ mg cm}^{-2}$ for each side and anode loads on two sides). 0.1 Ah pouch cells contain three layers of cathode and four layers of anode electrodes and constructed $\text{LiFePO}_4||\text{rGO/NbWO}$ pouch cells. In addition, the specific discharge capacity of $\text{LiFePO}_4||\text{rGO/NbWO}$ at 5 C was 85 mAh g^{-1} and remained 96.1% at the first 100 cycles. When the pouch cell was fast-charged and discharged for 500 cycles at 5 C, the capacity could still be maintained at around 77.0% (Fig. 5g). However, the current performance in practical applications is not ideal, primarily due to several limiting factors. As rGO/NbWO is a fast-charging anode material, most performance evaluations have been conducted with low loading levels ($1\sim 3 \text{ mg/cm}^2$). When the loading increases, its rate performance deteriorates significantly. For instance, the rGO/NbWO with an areal capacity of around 0.44 mAh cm^{-2} achieves 257 mAh g^{-1} at 0.2 C, when the loading is increased to 15 mg cm^{-2} with the areal capacity reaching 2.88 mAh cm^{-2} , the impedance rises to 198Ω , and the capacity is 208 mAh g^{-1} at 0.2 C and drops to 24 mAh g^{-1} at 40 C (Supplementary Fig. 28). Additionally, the commercial LFP shows a significant decline in rate performance under high current conditions, which results in the relatively low capacity of the $\text{LiFePO}_4||\text{rGO/NbWO}$ pouch cell (Supplementary Fig. 29). Fig. 5h

result shows that rGO/Nb₁₆W₅O₅₅ with a tap density of 1.61 g cm⁻³, can achieve an impressive energy density of 186 Wh kg⁻¹ at a high power density of 14,860 W kg⁻¹, and 406 Wh kg⁻¹ at 81 W kg⁻¹, super than most of NbWO based materials. Furthermore, using the BatPac model developed by Argonne National Laboratory^{50,51}, we simulated a 100 Ah LFP||rGO/ Nb₁₆W₅O₅₅ cell, which demonstrated a volumetric energy density of 251 Wh L⁻¹ and a gravimetric energy density of 123 Wh kg⁻¹, suggesting promising potential for practical applications.

Supplementary Fig. 28. The a) EIS and b) the rate performance of rGO/Nb₁₆W₅O₅₅ with different mass loading.

Supplementary Fig. 29. The a) rate performance and b) the cycling performance of the LFP cathode.”

8. Some statements should be rewritten for better clarity, e.g. “Graphite and silicon as LIB anode can store a large amount of Li⁺ ions within the potential range close to the Li⁺/Li redox couple; in practice, however, graphite stores Li⁺ at a low potential close to the Li plating reaction,....”. The way the statements are connected makes not too much

sense. Another example: “when the Li^+ ions that have already been desolvated from the electrolyte molecules get well prepared for their entry into the lattice..”

Response to comment-8: Thank you for your valuable suggestions. We carefully reviewed the language in the manuscript and have made further improvements for clarity and readability. The specific revisions on manuscript Page 2 and Page 15 are as follows:

“Graphite and silicon, both used as LIB anodes, can store significant amounts of Li^+ ions within a potential range close to the Li^+/Li redox couple^{6,7}. However, in practice, graphite stores Li^+ at a low potential close to the Li plating reaction, which can easily trigger lithium dendrite formation at high rates^{8,9}. While silicon offers a higher theoretical capacity, its potential for fast charging remains uncertain due to its low electronic conductivity and substantial volume changes during cycling¹⁰.” (On manuscript Page 2)

“This leads to the sluggish Li^+ intercalation around these shear planes, leading to the accumulation of more Li^+ at these surface regions, which could even be re-solvated into the electrolyte with thus severely compromised rate capability. As such, realizing that the anisotropic Li^+ transport property of the NbWO host acts as a major bottleneck toward approaching its ultrafast-charging milestone within 1 minute, we will discuss in the following how such anisotropy and its aftermath could be smartly addressed using a simple surface coating strategy.” (On manuscript Page 15)

9. Last sentence of the paper: “...pouch cells can charge mobile phones, and this development is expected to become the mainstream trend of future shared power”. I suggest removing such as statement. I do not understand its scientific value.

Response to comment-9: Thank you for your valuable suggestion. We have removed the statement in question and revised the conclusion for better clarity and focus on the scientific contribution of our work. The specific changes made on manuscript Page 24 are as follows:

“As a result, the rGO/NbWO exhibits a capacity of 257 mAh g^{-1} at 0.2 C and 116 mAh g^{-1} at 80 C, which maintains a capacity of 92.7% for 500 cycles at a rate of 10 C.

In addition, the $\text{LiFePO}_4\|\text{rGO}/\text{NbWO}$ pouch cell cycled at 5 C and remained 96.1% at the first 100 cycles and 77.0% capacity after 500 cycles at a large rate of 5 C. We anticipate that these direct, atomic-scale findings will deepen our understanding of the fast-charging mechanism of $\text{Nb}_{16}\text{W}_5\text{O}_{55}$ -based batteries.”

Point-by-point responses to reviewers' comments

(Blue type: Reviewer's comments; Black italic: Our response; Red type: Our revised)

Reviewer #1

The manuscript can be accepted as it has been clarified with all answers to review comments.

Response to Reviewer 1: Thank you for agreeing with acceptance.

Reviewer #2

The manuscript was revised appropriately, so it's now suitable for publication in Nat. Commun.

Response to Reviewer 2: We really appreciate your suggestions and support.

Reviewer #3

The manuscript clearly improved - again with a wealth of data. While this is useful and greatly acknowledged, the authors should simply discuss some of the data clearly in order to make the right claims. After these corrections, the paper could fulfill the standards of Nat. Comm..

Response to Reviewer 3: We sincerely appreciate the reviewer's thorough evaluation of our work. We have carefully addressed each of the reviewer's comments in the revised manuscript. Please find our detailed responses to your comments below.

Comment-1: "Supplementary Fig. 15. The FTIR spectrum of rGO/Nb₁₆W₅O₅₅." Needs a more critical discussion as the signal/noise is very weak. Also: What happens below 1250 cm⁻¹?

Response to comment-1: Thank you for your valuable comment. Since GO is reduced

to rGO at high temperatures, the C - OH and C=O signals on the material surface are weakened. Therefore, we have revised the statement to emphasize that “Compared to mechanically exfoliated graphene, reduced graphene oxide (rGO) boasts a simpler preparation method and contains a small number of functional groups, such as carboxyl and hydroxyl (Supplementary Fig. 15), which facilitate its integration with organic electrolytes in lithium-ion batteries.” This revision highlights the presence of limited functional groups on the rGO surface.

Additionally, the peaks observed below 1250 cm^{-1} , correspond to broad bands formed by numerous metal - oxygen - metal (M - O - M) bonds and/or M - O bonds. Specifically, the bands from 1100 to 800 cm^{-1} are associated with the short M - O bonds of the distorted MO_6 (M = Nb or W) octahedra within the crystallographic shear structures (*Energy Storage Materials*, 63, (2023) 102979).

The specific changes (On manuscript Page 18) and supporting information (on supporting information Page 13) include:

“Compared to mechanically exfoliated graphene, reduced graphene oxide (rGO) boasts a simpler preparation method and contains a small number of functional groups such as carboxyl and hydroxyl (Supplementary Fig. 15), facilitating its integration with organic electrolytes in lithium-ion batteries⁴⁰.”

Supplementary Fig. 15. The FTIR spectrum of rGO/Nb₁₆W₅O₅₅ (M = Nb or W).

Comment-2: Figure S20 at least contains a comparison.

Response to comment-2: Thank you for your insightful comment. We conducted two additional experiments comparing Nb₁₆W₅O₅₅ before and after rGO coating. The results show that the uncoated material does not exhibit D and G peaks for carbon, whereas the rGO-coated material displays a D-to-G peak ratio between 1.1 and 1.2. Based on these findings, we have made the following revisions to the manuscript (on the supporting information Page 17):

" Meanwhile, compared to NbWO, the Raman spectrum of rGO/NbWO shows distinct D and G peaks, with a D/G peak ratio between 0.988 and 1.053, indicating a higher degree of graphitization on the material's surface, which suggests improved electronic conductivity (Supplementary Fig. 20)."

Supplementary Fig. 20. The Raman spectra of Nb₁₆W₅O₅₅ and rGO/Nb₁₆W₅O₅₅ (each material was tested three times).

Comment-3: Figures S21: Details of the four probe method need to be mentioned so that other people can reproduce.

Response to comment-3: Thank you for your valuable feedback. The conductivity data were obtained from a four-probe method based on cryogenics probe station (Response Fig. 1) and tested by autolab 302N. As shown in Response Fig. 2, the probe spacing is denoted by s . A current is applied through probe 1, and the voltages across probes 2, 3, and 4 are measured. The resistivity equation can be simplified as $\rho = \frac{\Delta V_{23}}{2\pi s/I}$, where s is the spacing between two probes, I is the applied current, and ΔV_{23} is the voltage

difference between probes 2 and 3. The values ΔV_{23} and I are the measurement result from the autolab 302N, which is output directly by the testing instrument.

Response Fig. 1. Cryogenics probe station for I - V curves.

Response Fig. 2. Four-probe method Schematic diagram

Based on these findings, we have made the following revisions to the manuscript (on supporting information Page 17).

Supplementary Fig. 21. a) The I - V curves of $\text{Nb}_{16}\text{W}_5\text{O}_{55}$ and $\text{rGO}/\text{Nb}_{16}\text{W}_5\text{O}_{55}$ pressed pellets tested using the four-probe method based on cryogenics probe station and tested by autolab 302N. b) The resistivity equation can be simplified as $\rho = \Delta V_{23} 2\pi s / I$, where s is the spacing between two probes, I is the applied current, and ΔV_{23} is the voltage difference between probes 2 and 3. The values ΔV_{23} and I are the measurement result from the autolab 302N, which is output directly by the testing instrument.

Comment-4: Figure S22 needs to be replotted. There is hardly anything seen. Please provide information on the equivalent circuit and the quality of the fit. Indicate relevant frequency values. What was the frequency range of the experiment? Lot of things are unclear, experimental part should be more details on how the measurements were made. EIS data needs to be plotted in a x-y diagram with equal scaling of the axis.

Response to comment-4: Thank you for your detailed suggestions. We have replotted the EIS data, added the equivalent circuit, and provided the final simulated values. Additionally, we made revisions to the Supplementary Information and Experimental sections (on supporting information Pages 4 and 18), as follows:

“The *in situ* EIS performances of NbWO-based electrodes were investigated by an Autolab 302N at room temperature with a voltage range of 1 V to 3 V and the frequency range from 0.01 Hz to 100000 Hz.”

Supplementary Fig. 22. *In situ* EIS for a) rGO/Nb₁₆W₅O₅₅ and b) Nb₁₆W₅O₅₅ over a voltage range of 1 V to 3 V (frequency range: 0.01 Hz~100000 Hz). c) Equivalent circuit for fitting. d) Fitted values of impedance at different voltages.

Comment-5: I acknowledge that the authors used a large database to identify graphene as best material. Reduced graphene oxide, however, has not the same properties. Authors now state in the rebuttal “rGO also retains the electrochemical characteristics of conventional graphene stemming from its crystalline structure and size effects. Based on the above analysis, rGO was selected as a potential anode coating material ...”. This statement should be clearly reconsidered. For example, the oxidative stability of graphene will be very, very different from graphene oxide.

Response to comment-5: Thank you for your valuable comments. Graphene oxide (GO) is indeed very different from graphene. During the preparation process of reduced graphene oxide (rGO), graphene is first processed using strong oxidizing agents such as concentrated sulfuric acid and fuming nitric acid, followed by pyrolysis or ultrasonic dispersion for exfoliation. Subsequently, it undergoes reduction using reducing agents like hydrogen or potassium hydroxide to obtain reduced graphene oxide (rGO). During

the preparation of rGO, the process parameters for reduction, exfoliation, and oxidation are finely tuned to minimize the presence of extraneous organic functional groups on the graphene surface. However, it is still possible for trace amounts of unstable functional groups to remain on the rGO surface, resulting in a lower oxidation potential compared to pristine graphene, potentially lower than the predicted oxidation potential of 5.38 V for graphene. However, the operating potential of the anode in NbWO batteries ranges from 1~2.6 V vs Li/Li⁺. There is a large range between 5.38 V and 2.6 V, suggesting that the actual oxidation potential of rGO is highly likely to be above its upper operating limit of 2.6 V. Therefore, we still have sufficient reasons to believe that reduced graphene oxide (rGO) can serve as a stable coating material for NbWO anodes. To ensure accuracy, we have made modifications to the manuscript (On manuscript Page 19):

“The dataset after initial screening is presented in Fig. 4b, where each point represents a material with its x, y, and z coordinates signifying electronic conductivity (σ_e), ionic conductivity (σ_i), and oxidation potential (V_{ox}), respectively. Further refinement, targeting a V_{ox} above 4 V, σ_e above 10^3 S cm⁻¹, and σ_i above 10^{-6} S cm⁻¹, identifies graphene as the most promising coating material with the optimal comprehensive properties, as marked by the red dot in Fig. 4b. Specifically, its electronic conductivity, ionic conductivity, and oxidation potential are 9.2×10^3 S cm⁻¹, 5.61×10^{-6} S cm⁻¹, and 5.38 V, respectively. **Compared to mechanically exfoliated graphene, reduced graphene oxide (rGO) boasts a simpler preparation method and contains a small number of functional groups such as carboxyl and hydroxyl (Supplementary Fig. 15), facilitating its integration with organic electrolytes in lithium-ion batteries⁴⁰.** Based on the above analysis, rGO was selected as a potential anode coating material and conducted subsequent characterization and electrochemical experiments.”

Comment-6: Figure 4: Typo: Algorithm

Response to Comment-6: Thank you for pointing this out. We have corrected the typo

from "Algorithmim" to "Algorithm" as shown in the revised figure below (On manuscript Page 19):

Fig. 4. Fast charging performance enhancement. a The overall screening procedure from the Materials Project database. **b** Initial screening results of candidate coating materials under 3 selected dimensions: electronic conductivity, ionic conductivity, and oxidation potential.

Comment-7: Figure 4c: Should be improved. Y-axis has no numbers. Indicate the migration paths. I can not understand much from the illustration so far.

Response to Comment-7: Thank you for your valuable comment. We have added y-axis labels to Figure 4c and indicated the migration paths to enhance clarity. The revised figure now offers a more detailed and understandable illustration. Our DFT theoretical calculations (Fig. 4c) reveal that the diffusion energy barrier for lithium ions in rGO is lower than that within the NbWO tunnel structure. This suggests that coating NbWO with rGO can effectively enhance the material's overall ionic conductivity, thereby promoting significantly improved lithium-ion diffusion and enabling ultra-high rate performance. We have made modifications to the manuscript (On manuscript Page 19):

“Among various interface control strategies, the target rGO considering its low desolvation energy barrier and high lithium-ion diffusion rate that can guarantee fast Li^+ transport and relocation^{12,41}. DFT theoretical calculations (Fig. 4c) show that the diffusion energy barrier of lithium ions in rGO (I) is lower than that of migration and diffusion in NbWO tunnel structure (II~IV), which indicates that coating rGO in NbWO

can effectively increase the overall ionic conductivity of the material, which is conducive to the realization of significantly facilitated diffusion of lithium ions to obtain ultra-high rate performance.”

Fig. 4c Minimum energy paths and activation barriers (NEB) for lithium-ion motion in rGO/NbWO.

Comment-8: Figure 4d: Letter size in TEM images needs to be increased. It is not possible to read.

Response to Comment-8: Thank you for your suggestion. We have increased the font size in the TEM images in Figure 4d to improve readability. The updated figure now clearly displays all labels and details for easier interpretation. The specific revisions to the manuscript are as follows (On manuscript Page 19):

Fig. 4d I-V curves of NbWO and rGO/NbWO particles measured *in situ* under the microscope (scale bar 1 μm).

Comment-9: Energy density calculation: This is now clearer but it should be stated that

the values refer to the active materials only. In addition, authors used BatPac, which is good but information on what parameters were used is missing, e.g. what loading was assumed (0.44 mAh/cm²?). Maybe state the values in the supp.info? For the last sentence on the energy density: "...suggesting promising potential for practical applications", authors should compare to LTO/LFP data. If not, I would not make the claim but simply focus on the scientific discoveries of the paper. The low loading of 0.44 mAh/cm² seems not close to application.

Response to comment-9: Thank you for your insightful comments. We have indicated in the manuscript that the loading of 0.44 mAh cm⁻² is a conventional loading used specifically for coin cell testing. We increased the loading to 15 mg cm⁻², achieving an areal capacity of **2.88 mAh cm⁻²**. However, this higher loading is limited to manually prepared samples. Scaling up for industrial production would require significant process improvements, which would involve considerable time and effort. Therefore, we used the BatPaC simulation with an assumed areal capacity of approximately 3.8 mAh cm⁻² to estimate the performance potential. In comparison, the real capacity of conventional fast-charging LFP||LTO systems is only around 1.1 mAh cm⁻² (*Nat. Commun.* 15, 6299, 2024), indicating the promising potential of rGO/NbWO for practical applications. To ensure accuracy, we have made modifications to the manuscript (On manuscript Page 24):

“Furthermore, using the BatPac model developed by Argonne National Laboratory^{50,51}, we simulated a 100 Ah LFP||rGO/ Nb₁₆W₅O₅₅ cell with cathode and anode areal capacities reaching 3.8 mAh cm⁻². This simulation demonstrated a volumetric energy density of 251 Wh L⁻¹ and a gravimetric energy density of 123 Wh kg⁻¹.”

Comment-10: Figure 5: Scale coulombic efficiency more meaningful, e.g. from 80 – 110 %. When plotting it from 0-120% it always looks constant.

Response to comment-10: Thank you for your suggestion. We have adjusted the y-axis scale for coulombic efficiency in Figure 5 to a more meaningful range (e.g., 80% to 110%) to provide a clearer visualization of its variation. We have made modifications

to the manuscript (On manuscript Page 21):

Fig. 5. Fast charging performance evaluation. a Rate performance for rGO/NbWO and NbWO. **b** Comparison of rate performance in different works of literature. **c** Nyquist plot and fitting data for rGO/NbWO and NbWO. **d** dQ/dV plots of rGO/NbWO in 1 M LiTFSI. **e** Cycling performance of rGO/NbWO coin cells at 1 C and 10 C. **f** Schematic diagram of pouch cell. **g** Cycling performance of LFP||rGO/NbWO pouch cell at 5 C. **h** Energy density and power density of NbWO based materials.